# Model-Dowser: Data-Free Importance Probing to Mitigate Catastrophic Forgetting in Multimodal Large Language Models

Hyeontaek Hwang [* 1]   Nguyen Dinh Son [* 1]   Daeyoung Kim [1]

## Abstract

Fine-tuning Multimodal Large Language Models (MLLMs) on task-specific data is an effective way to improve performance on downstream applications. However, such adaptation often leads to a degradation in generalization on pretrained tasks, a phenomenon known as Catastrophic Forgetting. Existing methods that aim to mitigate this issue either become ineffective when fine-tuning deeper layers of the language decoder or scale poorly with increasing model size. To address these limitations, we propose Model-Dowser, a novel sparse fine-tuning approach for MLLMs. Model-Dowser measures a principled importance score for each model parameter with respect to pretrained generalization (prior to downstream adaptation) by jointly considering weight magnitudes, input activations, and output sensitivities. During fine-tuning, Model-Dowser selectively preserves high-importance parameters and updates the remaining. Comprehensive experiments on two representative MLLMs, LLaVA and NVILA, demonstrate that Model-Dowser effectively mitigates catastrophic forgetting and consistently outperforms prior methods, while remaining resource-efficient and scalable to multi-billion-parameter models. [1]

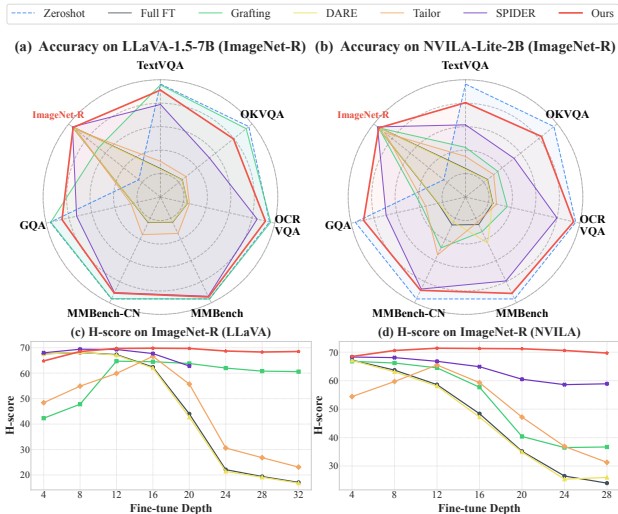

*Figure 1.* **Performance comparison of catastrophic forgetting mitigation methods on LLaVA-1.5 (7B) and NVILA-Lite (2B) fine-tuned on ImageNet-R. (a)-(b)** Radar charts illustrating the balance between downstream adaptation and upstream capabilities. **(c)-(d)** H-score stability across varying fine-tuning depths. **Model-Dowser** (red line) consistently achieves robust performance compared to previous works.

## 1. Introduction

Recently, Multimodal Large Language Models (MLLMs) (Liu et al., 2023; 2025; Wang et al., 2024a; Chen et al., 2024c) have emerged as a central focus in the field of multimodal learning. These models typically consist of a pretrained large language decoder (often a transformer-based architecture (Vaswani et al., 2017) integrated with vision encoders through a connection module. The success of MLLMs can be attributed to their ability to generalize across various tasks, which is driven by large-scale visual instruction tuning and their substantial model size, often reaching billions of parameters. Despite their strong zero-shot performance, these models frequently exhibit suboptimal results on domain-specific downstream tasks (Zhai et al., 2024; Zhu et al., 2024; Huang et al., 2025). As a result, further fine-tuning is often necessary to better align them with task-specific instructions and domain distributions.

A common approach to adapting MLLMs for specific tasks involves creating an instruction-following dataset tailored for the target task. During fine-tuning, the large language decoder is typically optimized while other components are kept frozen, enabling efficient alignment with downstream instructions (Han et al., 2024; Zhou et al., 2024a; Zhai et al., 2024). Although this fully supervised fine-tuning strategy is straightforward, it often leads to significant generaliza-

---

[*]Equal contribution [1]School of Computing, KAIST, Daejeon, Republic of Korea. Correspondence to: Hyeontaek Hwang <hwanght@kaist.ac.kr>, Nguyen Dinh Son < nguyendinhson@kaist.ac.kr>.

*Proceedings of the 43rd International Conference on Machine Learning*, Seoul, South Korea. PMLR 306, 2026. Copyright 2026 by the author(s).

[1]Code link: model-dowser.github.io

tion loss, a phenomenon known as *Catastrophic Forgetting* (Goodfellow et al., 2013; Zhai et al., 2024; Dong et al., 2021). This issue arises because downstream instruction data is usually limited in size and narrowly focused. As a result, fine-tuning MLLMs on such data can overwrite their pretrained representations.

Recent efforts to mitigate catastrophic forgetting in MLLMs mainly fall into two categories: *model post-merging* methods (Zhu et al., 2024; Yu et al., 2024; Panigrahi et al., 2023) and *sparse fine-tuning* methods (Huang et al., 2025; Hui et al., 2025). Post-merging approaches aim to preserve pretrained knowledge by fusing a pretrained model with its fine-tuned counterpart, whereas sparse fine-tuning methods restrict parameter updates to a subset of model weights to limit disruption to pretrained representations. Despite their effectiveness, existing approaches are predominantly evaluated under shallow fine-tuning settings, in which only a small subset of the language decoder's final layers is updated during downstream adaptation. For example, ModelTailor (Zhu et al., 2024) fine-tunes only the last 12 layers of LLaVA (Liu et al., 2023), while the state-of-the-art sparse method Specialization via Importance Discrepancy Evaluation for Refinement (SPIDER) (Huang et al., 2025) reports the results with less than the last 5 layers of LLaVA. However, recent studies (Chen et al., 2024b; Zhang et al., 2025) indicate that earlier layers of the language decoder play a critical role in multimodal understanding, suggesting that shallow fine-tuning may not fully exploit the model's adaptation capacity. Motivated by this observation, we examine catastrophic forgetting under deeper fine-tuning regimes, as illustrated in Figure 1. We find that post-merging methods degrade rapidly once fine-tuning extends to earlier decoder layers, likely because extensive parameter updates disrupt the pretrained latent space in ways that cannot be recovered through post-hoc fusion. Sparse fine-tuning methods exhibit more stable behavior in this setting, but their robustness comes at the expense of substantial memory and computational overhead, limiting scalability for MLLMs.

Driven by these limitations, we propose Model-Dowser, a novel sparse fine-tuning method for MLLMs that balances downstream task performance with preservation of pretrained generalization, while maintaining the same memory complexity as standard fine-tuning. Our approach is motivated by a simple question: ***Which parameter perturbations most strongly affect the model's outputs?*** We hypothesize that preserving generalization can be achieved by minimizing output shifts induced by downstream fine-tuning. To this end, we provide a theoretical analysis showing that output shifts induced by parameter updates can be effectively characterized by jointly considering weight magnitudes, input activations, and output sensitivities. Based on this insight, Model-Dowser assigns an importance score to each parameter prior to downstream adaptation and se-

lectively freezes high-importance parameters during fine-tuning, allowing the model to adapt to target tasks without overwriting pretrained knowledge.

Extensive experiments on two representative MLLM architectures, LLaVA (Liu et al., 2024a) and NVILA (Liu et al., 2025), across diverse downstream tasks demonstrate that Model-Dowser consistently achieves state-of-the-art performance in mitigating catastrophic forgetting. Furthermore, our method is data-free and computationally efficient, making it highly scalable to large models. Our contributions are summarized as follows:

- **Forgetting diagnosis across fine-tuning depth.** Our analysis reveals that MLLMs experience severe catastrophic forgetting as fine-tuning extends to deeper portions of the language decoder, and existing approaches are either ineffective under this regime or exhibit inconsistent behavior across different fine-tuning settings.

- **A scalable sparse fine-tuning method.** We propose Model-Dowser, a novel sparse fine-tuning method that introduces a data-free importance score derived from input activations and output sensitivity, and selectively freezes critical parameters prior to adaptation to enable effective downstream learning without loss of pretrained knowledge.

- **Theoretical justification.** We provide a theoretical analysis showing that our importance score captures the sensitivity of model outputs to individual parameter perturbations, explaining why preserving high-score parameters helps retain pretrained generalization.

- **State-of-the-art results with practical efficiency.** Our experiments on different MLLMs and downstream tasks show that Model-Dowser consistently outperforms prior methods, while remaining highly resource-efficient and scalable.

## 2. Related Works

### 2.1. Catastrophic Forgetting

In the context of large foundation models, catastrophic forgetting refers to the phenomenon where a model loses its ability to generalize to previously learned or unseen tasks after being adapted to downstream tasks (Wang et al., 2024b).

**In Large Language Models (LLMs)**, existing forgetting-mitigation methods can be broadly categorized into three groups. (1) *Additive methods* (Hu et al., 2022; Li & Liang, 2021; Lester et al., 2021; Zhang et al., 2021; Sung et al., 2022) introduce a small number of additional parameters to learn task-specific knowledge while keeping the pretrained weights fixed. Although these approaches are training-

efficient, the resulting architectural modifications complicate deployment and pose challenges for continual fine-tuning. (2) *Post-merging methods* (Yu et al., 2024; Panigrahi et al., 2023; Li et al., 2022) aim to fuse pretrained and fine-tuned weights using heuristic selection or importance criteria. However, such methods often struggle to balance generic knowledge and domain-specific adaptation, making them sensitive to hyperparameter choices and limiting their robustness. (3) *Sparse fine-tuning methods*, such as (Hui et al., 2025; Lu et al., 2024; Xu & Zhang, 2024), update a subset of model parameters during downstream adaptation and have shown potential in mitigating catastrophic forgetting. Nevertheless, due to the lack of a principled parameter importance criterion, these methods may inadvertently modify parameters critical to generalization.

**In Multimodal Large Language Models (MLLMs)**, early studies mainly investigate catastrophic forgetting as a side effect of continual learning (Guo et al., 2025a; Chen et al., 2025; 2024a; Guo et al., 2025b), which differs from our focus on downstream task adaptation. ModelTailor (Zhu et al., 2024) is among the first methods specifically designed for MLLMs, adopting a post-merging strategy. While it improves generalization compared to LLM-based baselines, it often sacrifices target-task performance to preserve pretrained representations. More recently, SPIDER (Huang et al., 2025) proposes a sparse fine-tuning approach that actively measures parameter importance during training based on gradient information and weight magnitudes, achieving strong empirical results in both downstream adaptation and generalization retention. However, existing MLLM-specific methods are typically evaluated only when fine-tuning the final layers of the language decoder, leaving their effectiveness under deeper fine-tuning largely unexplored. As we demonstrate later, these methods exhibit weak and unstable performance when fine-tuning extends to earlier layers.

We provide additional discussion of other anti-forgetting methods for smaller models in Appendix A and of continual learning in Appendix L.

### 2.2. Parameter Importance

Measuring parameter contributions is essential for mitigating catastrophic forgetting, yet early second-order methods such as Optimal Brain Surgeon (Hassibi & Stork, 1992) are computationally prohibitive for foundation models. As a result, magnitude-based methods (Mallya & Lazebnik, 2018; Tanaka et al., 2020) have emerged; however, they rely on assumptions of homogeneous activations, which do not hold in modern MLLMs. More recent approaches, including pruning by Weight AND Activation (Wanda) (Sun et al., 2024b), employ empirical criteria based on weight, activation, or gradient statistics, yet offer limited theoretical analysis of their impact on output-level functional stability. Moreover,

these criteria can be unreliable under massive activations (Sun et al., 2024a). SPIDER (Huang et al., 2025) employs dynamic updates during training based on weight magnitude and gradient norm, achieving balanced performance. However, SPIDER may incur substantial memory overhead due to maintaining the accumulated per-parameter gradient history and the soft mask matrix. To address these limitations, we propose a Model-Dowser that directly quantifies output-level functional impact, providing a principled and memory-efficient mechanism for preserving functionally critical parameters via a simple binary mask.

## 3. Methodology

In this section, we introduce Model-Dowser, which mitigates catastrophic forgetting by identifying and preserving functionally critical parameters through the three-stage pipeline illustrated in Figure 2. This process consists of data-free sensitivity probing and functional importance scoring to guide sparse fine-tuning and ensure stable representational adaptation.

### 3.1. Functional Importance Scoring

Modern MLLMs predominantly employ non-homogeneous activation functions, such as GELU (Hendrycks & Gimpel, 2016), SiLU (Elfwing et al., 2018), and GLU variants (Shazeer, 2020), in which the weight scale is no longer directly aligned with the output-level functional impact. In such architectures, the magnitude of a parameter no longer reliably reflects its functional contribution, as the non-linear curvature of such non-homogeneous activation can decouple weight scale from representational impact. To address this, we propose a sensitivity-based functional importance measure formalized in Theorem 3.1 to quantify importance via the estimated output shift $\|\Delta f\|_2$ induced by parameter perturbations. The corresponding proof is deferred to Appendix B.

**Theorem 3.1** (Functional shift for single-weight perturbation). *Consider a layer $l$ in an MLLM model $f$. Under first-order Taylor approximation, the L2 norm of output shift $\Delta f$ when perturbing a weight $W_{ij}^{(l)}$ is given by:*

$$\|\Delta f\|_2 \approx \|J_i^{(l)}\|_2 \cdot |\Delta W_{ij}^{(l)}| \cdot |h_j^{(l-1)}|, \qquad (1)$$

*where $J_i^{(l)} = \partial f / \partial z_i^{(l)}$ denotes the $i$-th column of the Jacobian matrix of the network output with respect to the pre-activation vector $z^{(l)}$, and $h^{(l-1)}$ is the input activation of the $l$-th layer (output of layer $l-1$).*

Theorem 3.1 can be extended to multi-weight perturbations, providing an upper bound under the first-order Taylor approximation for the total functional shift.

**Corollary 3.2** (Functional shift for multi-weight perturbation). *Let $\Delta \mathcal{W} = \{\Delta W^{(l)}\}_{l=1}^L$ be the set of perturbation*

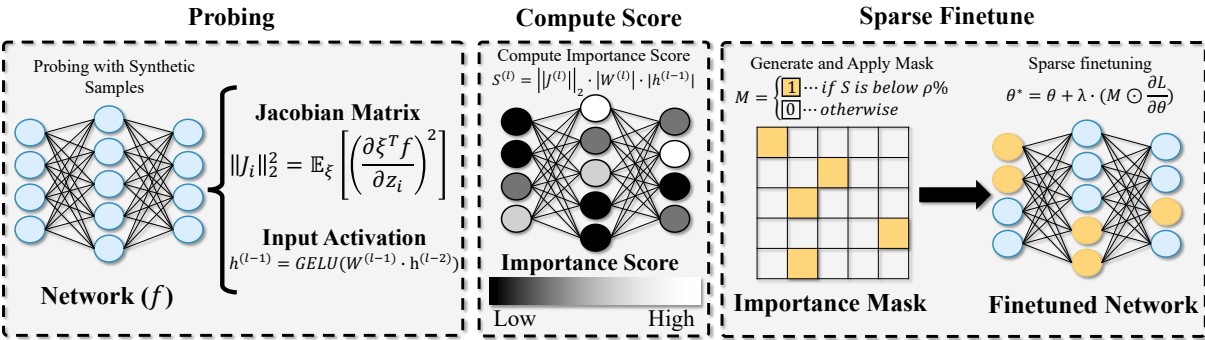

*Figure 2.* **Overall Architecture of Model-Dowser.** The proposed method consists of three main steps. **1. Probing** (Section 3.2): samples the Jacobian matrix and input activation with synthetic data samples on every layer ($l$). **2. Compute Score** (Section 3.2): generate parameter-wise importance score with Jacobian matrix, weight magnitude, and activation. **3. Sparse Finetune** (Section 3.3): update the least important $\rho\%$ of parameters (highlighted in yellow) based on their importance scores for the target downstream task.

*matrices for all layers in the model $f$. Under the first-order Taylor approximation, the total output shift $\|\Delta f\|_2$ is bounded by the global aggregate of individual parameter sensitivities as follows:*

$$\|\Delta f\|_2 \lessapprox \sum_{l,i,j} \|J_i^{(l)}\|_2 |\Delta W_{ij}^{(l)}| |h_j^{(l-1)}|. \tag{2}$$

The corresponding proof for this extension is deferred to Appendix C.

Based on the linear approximation provided in Theorem 3.1, we define the importance of each parameter. While the theorem quantifies the shift $\|\Delta f\|_2$ for an arbitrary perturbation $\Delta W$, we are specifically interested in the functional contribution of the current weight magnitude to the model's output stability. Following Theorem 3.1, we define a sensitivity-based importance score $S_{ij}^{(l)}$ by substituting the potential perturbation with the current weight magnitude $|W_{ij}^{(l)}|$:

$$S_{ij}^{(l)} = \|J_i^{(l)}\|_2 \cdot |W_{ij}^{(l)}| \cdot |h_j^{(l-1)}|. \tag{3}$$

We interpret $S_{ij}^{(l)}$ as a first-order proxy for the functional impact induced by the maximal local perturbation of $W_{ij}^{(l)}$. Intuitively, the score $S_{ij}^{(l)}$ quantifies the functional importance of a weight by considering the entire path of information flow in three dimensions:

**Output Sensitivity** ($\|J_i\|_2$) reflects the downstream impact, measuring the gradient-based sensitivity of the output to the $i$-th node.

**Connection Strength** ($|W_{ij}|$) represents the inherent scale of the parameter. As we interpret this as a proxy for $\Delta W_{ij}$, it captures the potential magnitude of the perturbation.

**Input Activity** ($|h_j|$) measures the magnitude of the in-coming signal from the preceding layer. It quantifies the connection's potential impact from input.

This formulation effectively integrates the gradient-based sensitivity of structural flow methods with the activation-awareness of local pruning, while providing robustness against the non-linear scaling of modern MLLM architectures.

### 3.2. Data-free Importance Estimation via Synthetic Probing

To compute the importance defined in Equation (3), we estimate each parameter's functional sensitivity by probing the model's response with synthetically generated inputs. This strategy avoids reliance on original pretraining data and enables scalable evaluation for large MLLMs.

**Stochastic Sensitivity Estimation** To capture the functional importance of each neuron, we utilize the L2 norm of the Jacobian $J$ of model $f$, which measures how much a perturbation at a specific node propagates to the final representation. However, computing the full Jacobian matrix of an MLLM is computationally expensive.

To calculate this efficiently in high-dimensional MLLMs, we utilize the Hutchinson Trace Estimator (Hutchinson, 1990). By projecting the output with a random Rademacher vector $\xi \in \{\pm 1\}^{d_{\text{final}}}$, the backward gain is stochastically estimated via the squared gradient:

$$\mathbb{E}_\xi \left[ \left( \frac{\partial(\xi^\top f)}{\partial z_i} \right)^2 \right] = \|J_i\|_2^2. \tag{4}$$

This allows us to obtain the node-wise output sensitivity of all parameters with a minimal number of backward passes, bypassing the need for explicit Jacobian construction. A discussion on the numerical stability for practice is in Ap-

pendix D.

**Synthetic Probing and Monte Carlo Estimation**    Finally, we address the challenge of data unavailability by performing the estimation with a synthetic probe. To enable data-free probing, we leverage MLLM's generative capability to synthesize $N$ prompts from random seeds, which serve as model-generated prompts to probe its functional response. The final importance score $\bar{S}_{ij}^{(l)}$ is computed as a Monte Carlo (MC) estimator, averaging the product of forward and backward gains over these $N$ stochastic trials:

$$\bar{S}_{ij}^{(l)} = \frac{1}{N} \sum_{n=1}^{N} \|J_{i,n}^{(l)}\|_2 \cdot |W_{ij}^{(l)}| \cdot |h_{j,n}^{(l-1)}|, \qquad (5)$$

where $J_{i,n}^{(l)}$ and $h_{j,n}^{(l)}$ are the Jacobian and activation values measured during the $n$-th trial. This averaging process serves as a variance reduction mechanism, ensuring that the identified functionally important parameters are robust across diverse activation patterns.

**Synthetic Prompt Generation for Data-Free probing**    In practice, pretraining data are often unavailable because they are in-house. To estimate parameter importance without access to such datasets, we leverage MLLMs' generative capabilities for data-free probing. Concretely, we sample random tokens from the model's tokenizer vocabulary and use them as seeds to synthesize $N$ prompts $\{\hat{x}_n\}_{n=1}^{N}$ as:

$$\hat{x}_n = f(\epsilon; \theta_{\text{pre}}), \qquad (6)$$

where $\epsilon$ is sampled token vector, and $\theta_{\text{pretrain}}$ is pretrained weight of a MLLM model. By propagating these synthetic probes through the model, we induce a wide range of activation patterns that reflect the model's learned functional structure, without relying on any task-specific or domain-specific data. This procedure enables us to identify structurally and functionally important parameters in a data-free manner. More detailed discussion and validation are provided in Appendix I.

**Computational Complexity**    The total procedure requires $O(N \cdot R)$ forward and backward passes, where $N$ is the number of MC synthetic samples and $R$ is the number of Rademacher vectors used for Hutchinson estimation per sample. Given that $N$ and $R$ are typically small (e.g., $N, R \ll d_{\text{final}}$), our method is significantly more efficient than explicit Jacobian computation, which would require $d_{\text{final}}$ backward passes.

### 3.3. Sparse Finetuning

Since catastrophic forgetting in MLLMs primarily manifests as functional drift of pretrained representations, preserving

parameters with high functional sensitivity naturally mitigates such degradation. Once the importance scores $\bar{S}_{ij}^{(l)}$ are estimated for all parameters, we perform sparse finetuning to adapt the MLLM to new tasks while mitigating catastrophic forgetting. This process involves two main steps: (1) identifying a set of functionally important parameters through binary masking and (2) performing gradient updates only on the non-essential parameters. To identify the most influential parameters for the pretrained weights, we rank all weights within each layer by their importance scores, $\bar{S}_{ij}^{(l)}$. Given an update ratio $\rho \in [0, 1]$, we generate a binary mask $M_{ij}^{(l)}$ as follows:

$$M_{ij}^{(l)} = \begin{cases} 1, & \text{if } \bar{S}_{ij}^{(l)} \text{ is in the bottom } \rho \text{ percentile,} \\ 0, & \text{otherwise.} \end{cases} \qquad (7)$$

During adaptation to a new task, we preserve knowledge encoded in functionally important weights by freezing them. The gradient updates are restricted to the remaining parameters (where $M_{ij}^{(l)} = 1$), allowing the model to learn task-specific features without shifting the established representation. The sparsely updated parameter $\theta^*$ during the finetuning is formulated as:

$$\theta^* = \theta - \lambda \cdot \left(M \odot \frac{\partial \mathcal{L}}{\partial \theta}\right), \qquad (8)$$

where $\theta$ denotes the weight of an MLLM model, $\lambda$ is learning rate, $\mathcal{L}$ is training objective, and $\odot$ represents the Hadamard product.

**Justification**    By freezing parameters with high $\bar{S}_{ij}^{(l)}$, we effectively suppress the dominant contributors to output perturbation, as approximated by the first-order sensitivity model in Theorem 3.1. Since our scoring mechanism accounts for the nonlinearity of non-homogeneous activations via the Hutchinson-estimated Jacobian, this sparse update strategy ensures that the functional anchors of the MLLM remain, providing greater stability than simple magnitude-based freezing methods.

## 4. Experiments

### 4.1. Experiment Settings

**Datasets and Architectures.**    To evaluate the effectiveness of our method, we fine-tune MLLMs on a diverse set of downstream tasks, including image captioning, image classification, and visual question answering (VQA). Specifically, we use COCO-Caption (Lin et al., 2014) and Flickr30k (Young et al., 2014) for image captioning, ImageNet-R (Hendrycks et al., 2021) for image classification, and IconQA (Lu et al., 2021) for VQA. Experiments are conducted on two representative MLLM architectures,

the widely used LLaVA (Liu et al., 2024a) and the recently proposed NVILA models (Liu et al., 2025). To assess generalization and forgetting, we follow prior benchmark protocols (Zhu et al., 2024; Huang et al., 2025) and evaluate zero-shot performance on upstream (pretrained) tasks using TextVQA (Singh et al., 2019), OKVQA (Marino et al., 2019), OCRVQA (Mishra et al., 2019), and GQA (Hudson & Manning, 2019). In addition, we include MMBench in both English (MMB) and Chinese (MMB (CN)) (Liu et al., 2024b) to measure multilingual zero-shot visual question answering capabilities. Please refer to Appendix E.1 for more details on each dataset.

**Baselines.** We compare Model-Dowser against several strong baselines designed to mitigate catastrophic forgetting. For methods specialized for MLLMs, we include ModelTailor (or **Tailor**) (Zhu et al., 2024) and **SPIDER** (Huang et al., 2025). ModelTailor is a post-merging approach that fuses pretrained weights with fine-tuned weights on downstream tasks, whereas SPIDER achieves state-of-the-art results in preventing forgetting for MLLMs by selectively updating model parameters based on accumulated gradient information during training. Given the architectural similarity between Large Language Models (LLMs) and Multimodal Large Language Models (MLLMs), we additionally consider representative forgetting-mitigation methods originally proposed for LLMs, including Model Grafting (**Grafting**) (Panigrahi et al., 2023) and Drop & Rescale (**DARE**) (Yu et al., 2024). Grafting and DARE are post-merging methods similar to ModelTailor. Finally, we report results from full fine-tuning (**Full-FT**) as a reference baseline. We provide more details on selected baselines in Appendix E.2.

**Training Settings.** We follow the resource settings from (Zhou et al., 2024b) and randomly sample 10k training instances from each downstream dataset. For all experiments, we use a learning rate of $2 \times 10^{-5}$ and train for 5 epochs. Unless otherwise specified, all remaining hyperparameters follow the official implementations of LLaVA[2] and NVILA[3]. Due to computational constraints, we evaluate LLaVA using the LLaVA-1.5-7B model and NVILA using NVILA-Lite-2B. During fine-tuning, we update the last $L$ layers of the language decoder while freezing all other components. To investigate the effect of fine-tuning depth, $L$ is varied in increments of 4, ranging from 4 to 32 for LLaVA and from 4 to 28 for NVILA. All experiments are conducted on 8 NVIDIA A100 GPUs (40 GB) with a total batch size of 128. For memory-intensive baselines such as Grafting and SPIDER, we instead use NVIDIA H200 GPUs (143 GB). SPIDER cannot be trained on LLaVA-1.5-7B when $L > 20$ due to its memory complexity. To calculate

the weight importance score in Model-Dowser, we set the number of synthetic text samples $N = 64$ and the number of Rademacher vectors $R = 8$ for all experiments; more details of these selections are in Appendix I.

**Evaluation Metrics.** For upstream (pretrained) tasks, we strictly follow the evaluation protocols of (Luo et al., 2024) and report the average accuracy across all upstream benchmarks, denoted as $A_{up}$, to measure the model's retained generalization ability. For downstream tasks, we adopt the CIDEr metric (Vedantam et al., 2015) for image captioning datasets (COCO-Caption and Flickr30k), and Exact Match(EM) accuracy for ImageNet-R and IconQA, denoted as $A_{down}$. We evaluate the effectiveness of forgetting mitigation using both the arithmetic mean (Avg) and harmonic mean (H-score) of $A_{up}$ and $A_{down}$:

$$\text{Avg} = \frac{A_{up} + A_{down}}{2}; \text{H-score} = \frac{2 \cdot A_{up} \cdot A_{down}}{A_{up} + A_{down}}. \quad (9)$$

Higher scores indicate better overall performance in mitigating catastrophic forgetting.

### 4.2. Findings

**Robust Balance between Upstream Knowledge and Downstream Adaptation.** Across all benchmarks, we observe a general trend: while different fine-tuning strategies often achieve comparable downstream performance, their ability to preserve upstream zero-shot knowledge varies. As shown in Tables 1 and 2, Model-Dowser achieves the highest H-scores among all evaluated methods by preserving pretrained knowledge during sparse fine-tuning. This trend holds across architectures. For instance, with NVILA-Lite-2B, Model-Dowser achieves H-scores of 85.7 and 71.2 on COCO-Caption and ImageNet-R, respectively, and also maintains strong upstream performance on LLaVA-1.5-7B while achieving H-scores of 79.9 and 69.7 on COCO-Caption and ImageNet-R, respectively. Importantly, these results suggest that catastrophic forgetting is less a consequence of insufficient downstream adaptation and more a result of failure to preserve functionally sensitive parameters. By selectively protecting parameters with high functional impact and updating those with minimal effect on the output function, Model-Dowser preserves the model's core representational capacity while retaining sufficient plasticity for downstream alignment, resulting in robust and balanced performance across models and tasks. We provide further details of experimental results in Appendix F

**Increased Vulnerability of Early Decoder Layers to Catastrophic Forgetting.** Layer-wise analysis shows that catastrophic forgetting is most severe when fine-tuning extends to early layers, as shown in Figure 3. Grafting, DARE, and Tailor maintain stability across the last 4–16 layers but

---

[2] https://github.com/haotian-liu/LLaVA
[3] https://github.com/NVlabs/VILA

*Table 1.* Performance comparison on COCO-Caption, ImageNet-R, Flickr30k, and IconQA using *NVILA-Lite 2B*, where only the last 20 layers of the language decoder are fine-tuned with an update ratio of $\rho = 0.1$. **Bold** and underlined entries represent the best and second-best results.

| Method | COCO-Caption | | | | | | | | | ImageNet-R | | | | | | | | |
|---|---|---|---|---|---|---|---|---|---|---|---|---|---|---|---|---|---|---|
| | TextVQA | OKVQA | OCRVQA | MMB | MMB(CN) | GQA | $A_{down}$ | Avg ↑ | H-Score ↑ | TextVQA | OKVQA | OCRVQA | MMB | MMB(CN) | GQA | $A_{down}$ | Avg ↑ | H-Score ↑ |
| Zeroshot | 70.8 | 51.2 | 69.4 | 78.1 | 42.2 | 62.3 | 26.1 | 44.2 | 36.8 | 70.8 | 51.2 | 69.4 | 78.1 | 42.2 | 62.3 | 22.2 | 42.2 | 32.7 |
| Full-FT | 8.1 | 11.5 | 65.1 | 25.1 | 15.5 | 24.0 | 98.5 | 61.7 | 39.7 | 16.0 | 10.8 | 13.3 | 43.0 | 33.1 | 14.2 | 92.3 | 57.0 | 35.2 |
| Grafting | 19.8 | 19.1 | 66.1 | 26.8 | 17.0 | 38.7 | 115.7 | 73.5 | 49.2 | 20.8 | 15.9 | 14.7 | 51.4 | 38.6 | 15.0 | 90.0 | 58.0 | 40.4 |
| DARE | 8.0 | 11.3 | 64.1 | 24.7 | 14.2 | 24.9 | 96.8 | 60.6 | 39.1 | 15.6 | 10.4 | 12.1 | 43.3 | 34.4 | 13.3 | 92.4 | 56.9 | 34.9 |
| Tailor | 7.7 | 12.5 | 64.2 | 40.7 | 25.9 | 18.9 | 105.6 | 67.0 | 44.7 | 28.8 | 18.7 | 23.9 | 60.2 | 42.5 | 26.2 | 80.3 | 66.9 | 47.2 |
| SPIDER | 65.3 | 42.3 | 67.8 | 72.0 | 48.4 | 59.6 | 115.4 | 87.3 | 78.3 | 42.5 | 26.1 | 54.3 | 68.0 | 38.9 | 40.9 | 91.9 | 68.5 | 60.5 |
| Dowser | 69.3 | 48.6 | 68.8 | 77.7 | 50.8 | 60.7 | **135.5** | **99.1** | **85.7** | 64.4 | 45.7 | 68.6 | 75.7 | 38.8 | 59.4 | 90.3 | **74.5** | **71.2** |
| | Flickr30k | | | | | | | | | IconQA | | | | | | | | |
| Zeroshot | 70.8 | 51.2 | 69.4 | 78.1 | 42.2 | 62.3 | 27.3 | 44.8 | 38.0 | 70.8 | 51.2 | 69.4 | 78.1 | 42.2 | 62.3 | 0.1 | 31.2 | 0.2 |
| Full-FT | 48.6 | 31.0 | 65.2 | 54.2 | 31.8 | 51.7 | 64.3 | 55.7 | 54.3 | 0.0 | 0.1 | 1.1 | 78.0 | 59.3 | 0.0 | 95.0 | 59.0 | 37.1 |
| Grafting | 60.3 | 37.9 | 64.9 | 61.9 | 39.3 | 55.8 | 76.8 | 65.1 | 63.0 | 0.0 | 0.0 | 0.2 | 77.3 | 59.6 | 0.0 | 94.5 | 58.7 | 36.8 |
| DARE | 48.5 | 30.6 | 64.9 | 52.7 | 30.2 | 50.9 | 63.9 | 55.1 | 53.7 | 0.0 | 0.0 | 0.9 | 77.9 | 59.3 | 0.0 | 95.0 | 59.0 | 37.1 |
| Tailor | 49.4 | 33.4 | 66.6 | 66.2 | 40.0 | 52.6 | 74.1 | 62.7 | 60.7 | 44.6 | 42.8 | 66.9 | 77.2 | 59.5 | 53.0 | 95.0 | 76.2 | 71.5 |
| SPIDER | 68.0 | 44.5 | 67.4 | 75.0 | 50.7 | 59.4 | 78.5 | 69.7 | 68.5 | 13.0 | 30.8 | 34.2 | 78.7 | 60.7 | 23.7 | 93.7 | 66.9 | 56.2 |
| Dowser | 70.1 | 48.6 | 68.0 | 77.9 | 51.0 | 60.3 | **96.0** | 79.3 | **75.8** | 71.2 | 50.7 | 69.0 | 77.8 | 57.7 | 62.2 | 92.0 | **78.4** | **76.0** |

*Table 2.* Performance comparison on COCO-Caption and ImageNet-R using *LLaVA-1.5-7B*, where only the last 20 layers of the language decoder are fine-tuned with an update ratio of $\rho = 0.1$. **Bold** and underlined entries represent the best and second-best results.

| Method | COCO-Caption | | | | | | | | | ImageNet-R | | | | | | | | |
|---|---|---|---|---|---|---|---|---|---|---|---|---|---|---|---|---|---|---|
| | TextVQA | OKVQA | OCRVQA | MMB | MMB(CN) | GQA | $A_{down}$ | Avg ↑ | H-Score ↑ | TextVQA | OKVQA | OCRVQA | MMB | MMB(CN) | GQA | $A_{down}$ | Avg ↑ | H-Score ↑ |
| Zeroshot | 58.3 | 58.0 | 65.1 | 64.6 | 58.2 | 62.0 | 40.3 | 50.7 | 48.5 | 58.3 | 58.0 | 65.1 | 64.6 | 58.2 | 62.0 | 16.3 | 38.7 | 25.8 |
| Full-FT | 52.7 | 48.8 | 63.0 | 63.9 | 58.0 | 57.5 | 100.0 | 78.7 | 72.9 | 24.3 | 10.2 | 26.2 | 48.6 | 45.0 | 20.6 | 89.7 | 59.4 | 44.0 |
| Grafting | 57.2 | 55.9 | 64.1 | 64.8 | 58.5 | 59.8 | 81.6 | 70.8 | 69.2 | 57.7 | 55.8 | 65.7 | 64.6 | 58.8 | 61.4 | 67.3 | 64.0 | 63.8 |
| DARE | 52.2 | 48.1 | 62.4 | 63.7 | 58.1 | 57.1 | 99.1 | 78.0 | 72.3 | 23.8 | 9.9 | 25.6 | 47.3 | 44.8 | 16.7 | 89.8 | 58.9 | 42.7 |
| Tailor | 55.0 | 50.9 | 64.5 | 64.8 | 58.4 | 58.8 | 106.7 | 82.7 | 75.8 | 38.1 | 18.2 | 40.1 | 61.4 | 54.8 | 36.8 | 84.3 | 62.9 | 55.7 |
| SPIDER | 56.1 | 53.2 | 64.7 | 63.9 | 57.5 | 59.8 | 103.1 | 81.1 | 75.2 | 46.8 | 27.4 | 55.6 | 63.0 | 54.8 | 43.0 | 89.3 | 68.9 | 62.8 |
| Dowser | 57.0 | 54.3 | 65.0 | 62.9 | 55.0 | 60.3 | **123.6** | **91.3** | **79.9** | 54.6 | 48.4 | 64.8 | 64.4 | 55.8 | 56.5 | 88.8 | **73.1** | **69.7** |
| | Flickr30k | | | | | | | | | IconQA | | | | | | | | |
| Zeroshot | 58.3 | 58.0 | 65.1 | 64.6 | 58.2 | 62.0 | 25.3 | 43.1 | 35.7 | 58.3 | 58.0 | 65.1 | 64.6 | 58.2 | 62.0 | 0.1 | 30.6 | 0.3 |
| Full-FT | 50.6 | 51.5 | 62.3 | 64.3 | 54.5 | 57.6 | 67.3 | 62.1 | 61.6 | 50.6 | 53.9 | 56.6 | 59.1 | 50.5 | 57.6 | 44.4 | 49.5 | 49.0 |
| Grafting | 56.9 | 55.9 | 64.1 | 64.9 | 58.2 | 60.4 | 64.0 | 62.0 | 62.0 | 58.1 | 58.2 | 65.1 | 63.4 | 56.3 | 61.8 | 22.4 | 41.4 | 32.7 |
| DARE | 55.3 | 51.4 | 62.8 | 64.5 | 54.6 | 58.3 | 66.3 | 62.1 | 61.8 | 50.4 | 53.8 | 56.2 | 59.1 | 50.5 | 57.0 | 44.5 | 49.5 | 49.0 |
| Tailor | 56.1 | 52.1 | 63.4 | 65.0 | 56.4 | 59.1 | 78.0 | 68.3 | 67.0 | 54.7 | 56.2 | 65.1 | 58.3 | 49.5 | 61.1 | 29.6 | 43.5 | 39.1 |
| SPIDER | 56.6 | 54.2 | 64.3 | 63.9 | 55.2 | 59.4 | 71.0 | 65.0 | 64.4 | 54.8 | 55.0 | 61.2 | 61.7 | 52.1 | 60.0 | 44.8 | 51.1 | 50.4 |
| Dowser | 56.5 | 55.2 | 63.3 | 63.8 | 55.1 | 59.6 | **85.0** | **72.0** | **69.6** | 57.7 | 58.5 | 65.3 | 63.0 | 53.9 | 61.8 | **44.9** | **52.4** | **51.3** |

collapse when updates are applied to early layers because these post-merging methods are inherently reactive; patching weights to recover the pretrained task is insufficient after disruption by fine-tuning. In contrast, SPIDER can preserve pretrained knowledge across even more layers by dynamically selecting parameters based on gradient history. However, even with all 32 layers, SPIDER underperforms Model-Dowser, indicating that it struggles to identify important parameters for preserving pretraining knowledge in the early layers. Conversely, Model-Dowser preserves sensitive functional anchors across layers and updates only the others, sustaining stability throughout the tuning process.

**Memory Efficiency and Scalability.** Model-Dowser provides a highly memory-efficient alternative to existing catastrophic forgetting mitigation strategies, maintaining the same complexity as standard fine-tuning. As shown in Table 3, while SPIDER and Grafting require significant

*Table 3.* Memory complexity for fine-tuning and parameter update ratios across different methods. $|P|$ denotes the number of learnable parameters. '# param' denotes the number of parameters on *NVILA/LLaVA* when updating the last 20 layers, respectively.

| Method | Memory Complexity | $\rho$ | # param |
|---|---|---|---|
| Full FT | $\mathcal{O}(|P|)$ | 100% | 1.4B / 4.5B |
| Grafting | $\mathcal{O}(2|P|)$ | 100% | 143M / 438M |
| DARE | $\mathcal{O}(|P|)$ | 10% | 143M / 438M |
| Tailor | $\mathcal{O}(|P|)$ | 10% | 143M / 438M |
| SPIDER | $\mathcal{O}(3|P|)$ | 50% | 714M / 2.3B |
| Dowser (Ours) | $\mathcal{O}(|P|)$ | 10% | 143M / 438M |

memory overheads of $\mathcal{O}(3|P|)$ and $\mathcal{O}(2|P|)$ respectively, Model-Dowser operates with a minimal memory complexity of $\mathcal{O}(|P|)$. This efficiency comes from Model-Dowser, which uses a static binary mask derived from a one-time calculation before fine-tuning; unlike SPIDER, which must

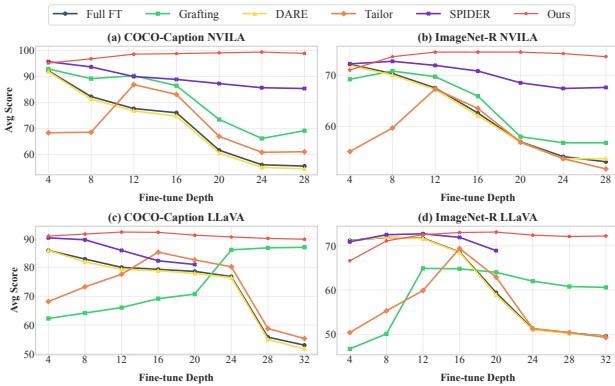

*Figure 3.* Performance comparison across fine-tuning depths on COCO and ImageNet-R. Results show the average accuracy across all tasks for an update ratio of $\rho = 0.1$ and various merging methods. The x-axis denotes the number of layers fine-tuned, counted incrementally from the final output layer toward the initial input layer.

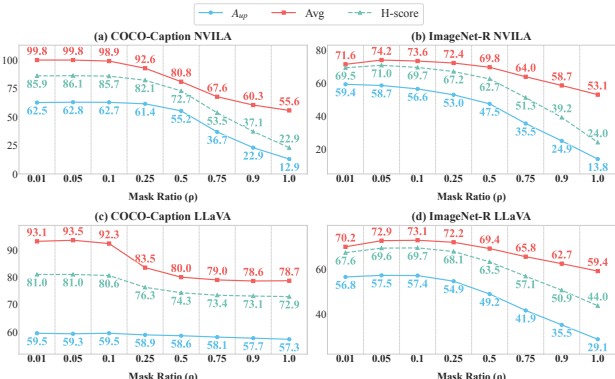

*Figure 4.* Performance comparison across various mask ratios ($\rho$) on COCO-Caption and ImageNet-R using *(a-b) NVILA-Lite-2b*, and *(c-d) LLaVA-1.5-7B*. Results show the upstream and downstream performance ($A_{up}$, Avg, H-score).

store and update accumulated gradient histories during training, our approach uses pre-computed importance scores to generate a fixed mask prior to fine-tuning. Because the cost of storing a binary mask is negligible, Model-Dowser can scale to large foundation models.

**Robustness to Update Ratios and the Stability.** Model-Dowser demonstrates a wide operational window regarding the update ratio $\rho$, effectively preserving stability even as the number of trainable parameters increases. In Figure 4, the average upstream performance remains stable for mask ratios up to $\rho = 0.25$, maintaining the upstream knowledge of the pretrained model. While a gradual degradation in upstream accuracy is observed as $\rho$ increases, Model-Dowser consistently outperforms the Full-FT ($\rho = 1.0$) across all evaluated settings. This empirical result suggests that the functional importance identified by our sensitivity scoring is highly concentrated; as long as the most critical parame-

*Table 4.* Result on ImageNet-R with different update ratios ($\rho$) across 28 layers on *NVILA-Lite-2B*. Random selection vs ours. Numbers after $\pm$ indicate standard deviation across three random seeds.

| Method | Ratio ($\rho$) | $A_{up}$ | $A_{down}$ | Avg ↑ | H-score ↑ |
|---|---|---|---|---|---|
| Full FT | 1.0 | 13.8 | 92.3 | 53.1 | 24.0 |
| Random | 0.1 | 54.5 $\pm 1.0$ | **90.7** $\pm 0.2$ | 72.6 $\pm 0.6$ | 68.0 $\pm 0.8$ |
| Ours | | **56.6** | 90.5 | **73.6** | **69.7** |
| Random | 0.25 | 49.0 $\pm 0.7$ | 91.5 $\pm 0.2$ | 70.2 $\pm 0.3$ | 63.8 $\pm 0.5$ |
| Ours | | **53.0** | **91.7** | **72.4** | **67.2** |
| Random | 0.5 | 39.2 $\pm 0.9$ | **92.2** $\pm 0.2$ | 65.7 $\pm 0.4$ | 55.0 $\pm 0.8$ |
| Ours | | **47.5** | 92.2 | **69.8** | **62.7** |
| Random | 0.75 | 25.6 $\pm 4.4$ | **92.6** $\pm 0.2$ | 59.1 $\pm 2.3$ | 40.1 $\pm 5.5$ |
| Ours | | **35.5** | 92.5 | **64.0** | **51.3** |
| Random | 0.9 | 20.4 $\pm 1.4$ | 92.6 $\pm 0.2$ | 56.5 $\pm 0.8$ | 33.4 $\pm 1.9$ |
| Ours | | **24.9** | 92.6 | **58.7** | **39.2** |

ters are protected, the model remains robust to significant task-specific updates in less sensitive regions. These results confirm that Model-Dowser provides a reliable mechanism for mitigating catastrophic forgetting without requiring exhaustive hyperparameter tuning for the mask ratio $\rho$.

**Comparing with Random Selection.** As shown in Table 4, Model-Dowser consistently outperforms the random selection (Xu & Zhang, 2024; Hui et al., 2025). The performance gap becomes most substantial at $\rho = 0.5$. Specifically, at $\rho = 0.5$, Model-Dowser achieves an Avg score of 69.8 and an H-score of 62.7, showing better performance over the random selection baseline ($65.7 \pm 0.4$ and $55.0 \pm 0.8$), respectively. Both gains are statistically significant ($p = 0.003$ for Avg and $p = 0.004$ for H-score under two-tailed t-tests).

The high variance observed under random selection reflects the instability of random updates, in which functionally sensitive parameters could be unintentionally modified. In contrast, by identifying these sensitive parameters, Model-Dowser ensures stability even when a substantial portion of the model is being updated. These results demonstrate that as the number of updated parameters increases, the reliability provided by importance-based parameter selection becomes essential for effectively mitigating forgetting. Further details are provided in Appendices F and I.

**Stability of the Data-Free Estimation on the Importance Score** We provide empirical evidence that our data-free importance estimation yields a stable, robust ranking of parameter sensitivity. We evaluate alignment with the real-data score from $N = 64$ text queries randomly sampled from the upstream tasks with $R = 8$ Rademacher vectors, using the Hamming distance (Hamming, 1950) and Spearman correlation (SPEARMAN, 1904). As shown in Table 5,

*Table 5.* **Comparison of parameter importance** Hamming distance and Spearman correlation relative to the score with real-data samples. Results are presented as 'mean $\pm$ std' over 5 independent runs. $R$ and $N$ denote the number of Rademachers and Monte Carlo samples, respectively.

| R | N | Hamming Distance ($\downarrow$) | | Spearman Correlation ($\uparrow$) | |
|---|---|---|---|---|---|
| | | Random | Ours | Random | Ours |
| 2 | 4 | $0.226 \pm 0.000$ | $\mathbf{0.064 \pm 0.004}$ | $0.000 \pm 0.000$ | $\mathbf{0.861 \pm 0.008}$ |
| | 64 | $0.226 \pm 0.000$ | $\mathbf{0.052 \pm 0.002}$ | $0.000 \pm 0.000$ | $\mathbf{0.886 \pm 0.004}$ |
| 8 | 4 | $0.226 \pm 0.000$ | $\mathbf{0.058 \pm 0.003}$ | $0.000 \pm 0.000$ | $\mathbf{0.875 \pm 0.007}$ |
| | 64 | $0.226 \pm 0.000$ | $\mathbf{0.050 \pm 0.002}$ | $0.000 \pm 0.000$ | $\mathbf{0.891 \pm 0.003}$ |

our estimated importance score closely aligns with real-data ranking as N increased, whereas random selection shows no alignment. With $R = 8, N = 64$, Model-Dowser achieves a Hamming distance of $0.050 \pm 0.002$ and Spearman correlation of $0.891 \pm 0.003$, indicating that our data-free estimation effectively captures the model's sensitivity. We provide detailed analysis, including text-only probing in Appendices I and J.

## 5. Conclusion

In this paper, we investigate catastrophic forgetting in MLLMs under varying fine-tuning depths. Our analysis reveals that fine-tuning earlier layers of the language decoder severely degrades pretrained generalization, and existing methods often fail and exhibit unstable performance. To address this challenge, we propose Model-Dowser, which identifies the parameters most impactful on the model's outputs prior to downstream adaptation and selectively preserves them during fine-tuning. As a result, Model-Dowser effectively mitigates catastrophic forgetting while remaining data-free and resource-efficient, requiring no additional memory beyond standard fine-tuning and thus scaling well to large MLLMs.

## Acknowledgement

This work was supported by the National Research Foundation of Korea (NRF) grant funded by the Korea government (MSIT) (RS-2025-00573160), the IITP (Institute of Information & Communications Technology Planning & Evaluation)-ITRC (Information Technology Research Center) grant funded by the Korea government (Ministry of Science and ICT) (IITP-2026-RS-2023-00259703), and the "Advanced GPU Utilization Support Program" funded by the Government of the Republic of Korea (Ministry of Science and ICT).

## Impact Statement

This paper presents work whose goal is to advance the field of Machine Learning. There are many potential societal consequences of our work, none which we feel must be specifically highlighted here.

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

# Appendix of Model-Dowser: Data-Free Importance Probing to Mitigate Catastrophic Forgetting in MLLMs

In this appendix, we provide a more detailed discussion, proof, and result from the main text.

- **Appendix A**: An extended discussion on the related work from Section 2.

- **Appendix B**: Proof of Theorem 3.1 (cf. Section 3.1).

- **Appendix C**: Proof of Corollary 3.2 (cf. Section 3.1).

- **Appendix D**: $L_1$-norm Surrogate and corresponding proof.

- **Appendix E**: Detailed setting of experiments and definition of metrics (cf. Section 4.1).

- **Appendix F**: Detailed experiments on various benchmarks and further analysis on the proposed importance score.

- **Appendix G**: Effect of learning rate on upstream retention and downstream adaptation across methods.

- **Appendix H**: Comparison with random selection and importance-score-based mask.

- **Appendix I**: Stability analysis on the proposed data-free estimation method.

- **Appendix J**: Validation of text-only probing by comparing probing with image input.

- **Appendix K**: Ablation on Jacobian ($\|J\|_2$) term

- **Appendix L**: Additional experiment on Continual Learning setting.

## A. More discussion of Related Work

Catastrophic forgetting is a long-standing challenge in deep neural networks. Early studies primarily investigated this phenomenon in unimodal models under continual learning settings, where a model is sequentially trained on new tasks, and its performance on previously learned tasks degrades (Goodfellow et al., 2013; Masana et al., 2022; Yang et al., 2023; Kirkpatrick et al., 2017; Xuhong et al., 2018; Aljundi et al., 2018; Zhang et al., 2024b). These works typically assume a single-source task constraint, in which successive tasks are closely related, e.g., image classification, and the source-task training data remains accessible during adaptation.

In contrast, the emergence of foundation models (Radford et al., 2021; Kirillov et al., 2023; Zhai et al., 2023) introduces a fundamentally different form of forgetting. In this setting, the original pretraining data distribution is often unknown or unavailable, and catastrophic forgetting manifests as a loss of generalization after fine-tuning on a downstream task. Recent efforts to mitigate forgetting in foundation models have largely focused on gradient-based or full-model fine-tuning methods (Zhang et al., 2024a; Zheng et al., 2023; Xiang et al., 2023); however, such approaches are often impractical for large-scale pretrained models due to their computational and memory overhead.

As Multimodal Large Language Models (MLLMs) (Liu et al., 2024a; 2025; Wang et al., 2024a) continue to demonstrate strong adaptability across diverse downstream tasks, understanding and mitigating catastrophic forgetting in these models has become an increasingly important research direction.

## B. Functional shift for single-weight perturbation (Proof of Theorem 3.1)

**Theorem B.1** (Theorem 3.1). *Consider a layer $l$ in an MLLM model $f$. Under first-order Taylor approximation, the L2 norm of output shift $\Delta f$ when perturbing a weight $W_{ij}^{(l)}$ is given by:*

$$\|\Delta f\|_2 \approx \|J_i^{(l)}\|_2 \cdot |\Delta W_{ij}^{(l)}| \cdot |h_j^{(l-1)}|, \tag{10}$$

*where $J_i^{(l)} = \partial f / \partial z_i^{(l)}$ denotes the $i$-th column of the Jacobian matrix of the network output with respect to the pre-activation vector $z^{(l)}$, and $h^{(l-1)}$ is the input activation of the $l$-th layer (output of layer $l-1$).*

*Proof.* In this section, we provide the detailed derivation of the parameter importance score $S_{ij}^{(l)}$ based on the first-order Taylor approximation of the model's output shift.

**Setup and Notation**   Consider a specific linear layer $l$ within an MLLM $f$. For a given input stimulus, let $h^{(l-1)} \in \mathbb{R}^{d_{\text{in}}}$ denote the input activation (output from the preceding layer) and $W^{(l)} \in \mathbb{R}^{d_{\text{out}} \times d_{\text{in}}}$ denote the weight matrix of layer $l$. The pre-activation vector $z^{(l)}$ is defined as $z^{(l)} = W^{(l)} h^{(l-1)}$, where the $i$-th component is $z_i^{(l)} = \sum_j W_{ij}^{(l)} h_j^{(l-1)}$. The network output $f$ can be viewed as a function of the pre-activation $z^{(l)}$, denoted as $f = F(z^{(l)})$, where $F$ encompasses all subsequent operations in the network. We define the output sensitivity vector (Jacobian column) $J_i^{(l)}$ as:

$$J_i^{(l)} = \frac{\partial f}{\partial z_i^{(l)}} \in \mathbb{R}^{d_{\text{final}}}. \tag{11}$$

**Derivation for Single Weight Perturbation**   We analyze the effect of perturbing a single weight element $W_{ij}^{(l)}$ by an amount $\Delta W_{ij}^{(l)}$.

**Step 1:** Since only the weight element $W_{ij}^{(l)}$ is perturbed, only the $i$-th component of the pre-activation vector $z^{(l)}$ is affected:

$$z_i'^{(l)} = \sum_k (W_{ik}^{(l)} + \delta_{kj} \Delta W_{ij}^{(l)}) h_k^{(l-1)} = z_i^{(l)} + \Delta W_{ij}^{(l)} h_j^{(l-1)}, \tag{12}$$

where $\delta$ is the Kronecker delta. Thus, the change in the $i$-th pre-activation is $\Delta z_i^{(l)} = h_j^{(l-1)} \Delta W_{ij}^{(l)}$, while $\Delta z_k^{(l)} = 0$ for all $k \neq i$.

**Step 2:** To quantify the functional impact of a parameter perturbation, we treat the final network output $f$ as a differentiable function of the pre-activation $z^{(l)}$. When the pre-activation vector is perturbed from $z^{(l)}$ to $z^{(l)} + \Delta z^{(l)}$, we can approximate the new output using a first-order Taylor expansion around $z^{(l)}$:

$$F(z^{(l)} + \Delta z^{(l)}) = F(z^{(l)}) + \frac{\partial F}{\partial z^{(l)}} \Delta z^{(l)} + \mathcal{O}(\|\Delta z^{(l)}\|^2) \tag{13}$$

By neglecting the higher-order terms $\mathcal{O}(\|\Delta z^{(l)}\|^2)$ and rearranging the equation to isolate the difference between the perturbed and original outputs, we define the output shift $\Delta f$ as follows:

$$\Delta f = F(z^{(l)} + \Delta z^{(l)}) - F(z^{(l)}) \approx \frac{\partial F}{\partial z^{(l)}} \Delta z^{(l)} = \sum_{k=1}^{d_{\text{out}}} J_k^{(l)} \Delta z_k^{(l)}, \tag{14}$$

where $\frac{\partial F}{\partial z^{(l)}} = [J_1^{(l)}, J_2^{(l)}, \ldots, J_{d_{\text{out}}}^{(l)}]$ represents the Jacobian matrix of the network output with respect to the pre-activations. As established in Step 1, since the perturbation is restricted to a single weight $W_{ij}^{(l)}$, the change vector $\Delta z^{(l)}$ is sparse, containing a non-zero value only at the $i$-th index ($\Delta z_i^{(l)} = \Delta W_{ij}^{(l)} \cdot h_j^{(l-1)}$). Consequently, the summation collapses to a single term:

$$\Delta f \approx J_i^{(l)} \Delta z_i^{(l)} = J_i^{(l)} \cdot (\Delta W_{ij}^{(l)} \cdot h_j^{(l-1)}). \tag{15}$$

**Step 3:** Taking the $L_2$ norm of both sides, we obtain the magnitude of the functional shift:

$$\|\Delta f\|_2 \approx \|J_i^{(l)} \cdot (\Delta W_{ij}^{(l)} \cdot h_j^{(l-1)})\|_2 = \|J_i^{(l)}\|_2 \cdot |\Delta W_{ij}^{(l)}| \cdot |h_j^{(l-1)}|. \tag{16}$$

By substituting the potential perturbation $\Delta W_{ij}^{(l)}$ with the existing weight magnitude $|W_{ij}^{(l)}|$, we arrive at the importance score $S_{ij}^{(l)} = \|J_i^{(l)}\|_2 \cdot |W_{ij}^{(l)}| \cdot |h_j^{(l-1)}|$. We emphasize that this substitution does not aim to predict the exact output shift, but serves as a conservative first-order surrogate for ranking parameters by their relative functional sensitivity.

The proof of Theorem 3.1 is finished. □

## C. Extension to Multi-layer and Multi-weight Pruning (Proof of Corollary 3.2)

**Corollary C.1** (Corollary 3.2). *Let $\Delta \mathcal{W} = \{\Delta W^{(l)}\}_{l=1}^L$ be the set of perturbation matrices for all layers in the model $f$. Under the first-order Taylor approximation, the total output shift $\|\Delta f\|_2$ is bounded by the global aggregate of individual parameter sensitivities as follows:*

$$\|\Delta f\|_2 \lesssim \sum_{l,i,j} \|J_i^{(l)}\|_2 \cdot |\Delta W_{ij}^{(l)}| \cdot |h_j^{(l-1)}|. \tag{17}$$

*Proof.* We derive this using the concept of the total differential, treating the neural network function $f$ as differentiable with respect to the set of parameters $\mathcal{W} = \{W_{ij}^{(l)}\}_{l,i,j}$.

According to the definition of the total differential, the variation in the output $\Delta f$ induced by simultaneous perturbations in all weights can be approximated by the sum of partial derivatives with respect to every weight parameter:

$$\Delta f \approx \sum_{l=1}^L \sum_{i,j} \frac{\partial f}{\partial W_{ij}^{(l)}} \Delta W_{ij}^{(l)}. \tag{18}$$

From the derivation in Theorem 3.1 (Theorem B.1), we established that the gradient of $f$ with respect to a specific weight $W_{ij}^{(l)}$ factorizes via the chain rule into the downstream sensitivity and the input activation:

$$\frac{\partial f}{\partial W_{ij}^{(l)}} = \frac{\partial f}{\partial z_i^{(l)}} \cdot \frac{\partial z_i^{(l)}}{\partial W_{ij}^{(l)}} = J_i^{(l)} \cdot h_j^{(l-1)}. \tag{19}$$

Substituting this result directly into Equation (18) yields:

$$\Delta f \approx \sum_{l=1}^L \sum_{i,j} \left( J_i^{(l)} \cdot h_j^{(l-1)} \right) \Delta W_{ij}^{(l)}. \tag{20}$$

Finally, to bound the magnitude of the total shift, we apply the triangle inequality ($\|\sum \mathbf{x}\| \leq \sum \|\mathbf{x}\|$) and the property of scalar multiplication:

$$\|\Delta f\|_2 \approx \left\| \sum_{l,i,j} J_i^{(l)} h_j^{(l-1)} \Delta W_{ij}^{(l)} \right\|_2 \leq \sum_{l,i,j} \|J_i^{(l)}\|_2 \cdot |h_j^{(l-1)}| \cdot |\Delta W_{ij}^{(l)}|. \tag{21}$$

$$\therefore \|\Delta f\|_2 \lesssim \sum_{l,i,j} \|J_i^{(l)}\|_2 \cdot |\Delta W_{ij}^{(l)}| \cdot |h_j^{(l-1)}|. \tag{22}$$

This inequality demonstrates that the sum of our proposed importance scores, $S_{ij}^{(l)}$, serves as a theoretical upper bound on the total functional shift of the MLLM under first-order approximation. Consequently, minimizing this aggregate score via global parameter selection effectively preserves the pretrained model's established functional response under perturbations.

The proof of Corollary 3.2 is finished. □

# D. $L_1$-norm Surrogate for Scalable Parameter Importance Estimation

**Theorem D.1.** *Consider a MLLM $f \in \mathbb{R}^{d_{final}}$ and a Rademacher random variable $\xi \in \{-1, 1\}^{d_{final}}$. Under the sensitivity measure defined in Theorem 3.1, the following inequality holds:*

$$\sqrt{\tfrac{1}{2}} \, \|J_i\|_2 \cdot |\Delta W_{ij}| \cdot |h_j| \ \leq \ \mathbb{E}_\xi[|\xi^\top J_i|] \cdot |\Delta W_{ij}| \cdot |h_j| \ \leq \ \|J_i\|_2 \cdot |\Delta W_{ij}| \cdot |h_j|, \tag{23}$$

*where $J_i = \partial f / \partial z_i$ denotes the $i$-th column vector of the Jacobian matrix, $h$ is the activation, and $W$ is the weight matrix of the model $f$.*

*Proof.* First, we recall that the $L_2$-norm of the Jacobian matrix can be estimated using the Hutchinson trace estimator. For a given component or vector $J$, the relationship is established as:

$$\|J_i\|_2^2 = \mathbb{E}_\xi \left[ \left( \frac{\partial(\xi^\top f)}{\partial z_i} \right)^2 \right] = \mathbb{E}_\xi[(\xi^\top J_i)^2]. \tag{24}$$

To relate the expectation of the absolute value to the $L_2$-norm, we first apply Jensen's inequality for the concave function $\sqrt{\cdot}$, which yields:

$$\mathbb{E}_\xi[|\xi^\top J_i|] \leq \sqrt{\mathbb{E}_\xi[(\xi^\top J_i)^2]} = \sqrt{\|J_i\|_2^2}. \tag{25}$$

This provides the upper bound $\mathbb{E}_\xi[|\xi^\top J_i|] \leq \|J_i\|_2$.

Furthermore, to establish the lower bound, we utilize the Khintchine inequality (Khintchine, 1923; Haagerup, 1981). For the specific case of $p = 1$ and Rademacher complexity, the inequality states:

$$\sqrt{\tfrac{1}{2}} \, \|J_i\|_2 \ \leq \ \mathbb{E}_\xi[|\xi^\top J_i|] \ \leq \ \|J_i\|_2. \tag{26}$$

This relationship demonstrates that the expectation $\mathbb{E}_\xi[|\xi^\top J_i|]$ is equivalent to the $L_2$-norm of the Jacobian up to a constant factor, i.e., $\mathbb{E}_\xi[|\xi^\top J_i|] \asymp \|J_i\|_2$. Given that $\|J_i\|_2 \leq \|J_i\|_1$, this estimator serves as a robust proxy for the sensitivity of the model outputs.

Finally, by multiplying all terms by the magnitude of the activation $|h_j|$ and the weight perturbation $|\Delta W_{ij}|$, we obtain the following inequality:

$$\sqrt{\tfrac{1}{2}} \, \|J_i\|_2 \cdot |\Delta W_{ij}| \cdot |h_j| \ \leq \ \mathbb{E}_\xi[|\xi^\top J_i|] \cdot |\Delta W_{ij}| \cdot |h_j| \ \leq \ \|J_i\|_2 \cdot |\Delta W_{ij}| \cdot |h_j|. \tag{27}$$

The proof of Theorem D.1 is finished. □

**Numerical Stability near Zero.**    A challenge in using the $L_2$-norm with low-precision formats (e.g., FP16 or BF16) is the risk of underflow. When Jacobian elements $J_i$ are small, the squaring operation in $\|J\|_2$ can push values below the minimum representable threshold, causing them to be flushed to zero. This results in a loss of the relative-importance ranking among parameters.

Our $L_1$-based estimator $\mathbb{E}_\xi[|\xi^\top J_i|]$ avoids this by preserving the original magnitude of the gradients, ensuring that even subtle sensitivity differences are captured without numerical vanishing. In this proof, we show that, in practice, the proposed estimator and the $L_2$ norm have equivalent growth rates. Specifically, since the estimator scales linearly with $\|J_i\|_2$, both methods can serve as effective surrogates for the parameter importance score without losing the relative ranking of sensitivity.

# E. Experiment setting details.

## E.1. Details of Datasets

**TextVQA** (Singh et al., 2019) is a visual question answering benchmark that requires models to read and reason about text embedded in images to answer natural language questions. It evaluates a model's ability to jointly perform scene text understanding and visual–language reasoning, highlighting limitations of standard VQA models that lack text-reading capability.

**OKVQA** (Marino et al., 2019) requires models to answer questions by combining visual understanding with external, commonsense, or factual knowledge beyond what is directly observable in the image. It is designed to evaluate a model's ability to integrate vision with knowledge-based reasoning.

**OCR-VQA** (Mishra et al., 2019) focuses on answering questions by reading and understanding text present in images. It evaluates a model's ability to perform optical character recognition and associate recognized text with visual context to support accurate question answering.

**MMBench** (Liu et al., 2024b) is a comprehensive multimodal benchmark designed to evaluate the general capabilities of multimodal models across diverse vision–language skills, including perception, reasoning, and knowledge understanding. It provides standardized multiple-choice questions split between English and Chinese, enabling systematic evaluation of zero-shot and multilingual performance in multimodal large language models.

**GQA** (Hudson & Manning, 2019) is a visual question answering benchmark designed to evaluate real-world visual reasoning and compositional understanding. It features structured questions that require multi-step reasoning over objects, attributes, and relations in images, enabling fine-grained analysis of a model's visual reasoning capabilities.

**COCO-Caption** (Lin et al., 2014) is a large-scale vision dataset containing images of complex everyday scenes with multiple objects annotated for object detection, segmentation, and image captioning. It is widely used to evaluate visual understanding and language generation capabilities, particularly in image captioning and multimodal learning tasks.

**Flickr30k** (Young et al., 2014) is an image–caption dataset consisting of 30,000 everyday images, each annotated with multiple human-written captions. It is commonly used to evaluate image captioning and vision–language understanding by assessing a model's ability to generate descriptive, semantically accurate natural-language captions.

**IconQA** (Lu et al., 2021) is a visual question answering benchmark designed for abstract diagram understanding and visual–language reasoning. In this paper, we focus on the text-based multiple-choice split.

**ImageNet-R** (Hendrycks et al., 2021) is a robustness benchmark derived from ImageNet that contains images rendered in diverse artistic styles, such as paintings, cartoons, and sketches. It is designed to evaluate a model's out-of-distribution generalization by testing recognition performance under significant appearance shifts from natural images. In this paper, we adopt the visual question answering format of this dataset from (Guo et al., 2025a).

## E.2. Details of Baselines

**Model Grafting** (Panigrahi et al., 2023) optimizes the model's loss on the downstream task with respect to a merging mask, which is later used to fuse the pretrained model and the fine-tuned model. While additional mask training can maintain the merged model's performance on the target task, it cannot guarantee performance on pretrained tasks. Moreover, this training requires $2\times$ the memory compared to standard fine-tuning, thus limiting its applicability to large models.

**DARE** (Yu et al., 2024) aims to reduce the delta weights without additional training. It first randomly drops a proportion of delta weights, then rescales the remaining ones to eliminate redundant parameters. The rescaled delta weights are later added back to the pretrained models. Despite its simplicity, DARE shows suboptimal results and inconsistent performance when applied to MLLMs.

**ModelTailor** (Zhu et al., 2024) is the first work that explores the catastrophic forgetting problem in MLLMs. ModelTailor proposed a post-training fusion strategy based on salience and sensitivity analysis. Although it is specifically designed for MLLMs, ModelTailor still suffers from severe forgetting of pretrained knowledge when fine-tuning a deeper proportion of language layers, similar to other post-merging methods.

**SPIDER** (Huang et al., 2025) is the state-of-the-art method for MLLMs, which adopts sparse-finetuning instead of post-merging strategies. By actively monitoring the accumulated parameter gradients and magnitudes, SPIDER ranks the

parameter importance and selectively updates a subset of parameters. While this approach demonstrates better results compared to previous methods, it introduces substantial computational overhead and memory usage ($3\times$ compared to the standard fine-tuning).

# F. Further Experiments

## F.1. Further Details on Experimental Result

As shown in Figure 5, Model-Dowser differs from previous approaches by remaining stable even when fine-tuning is applied across the full model, while the baselines show a performance drop. This aligns with the analysis in Section 4, suggesting that our method precisely targets parameters crucial for balancing knowledge retention and task adaptation. Additionally, Figure 6 highlights the method's resilience to different mask ratios ($\rho$); specifically, the consistent $A_{up}$ scores across all settings attest to the efficacy of our importance score in preserving upstream capabilities.

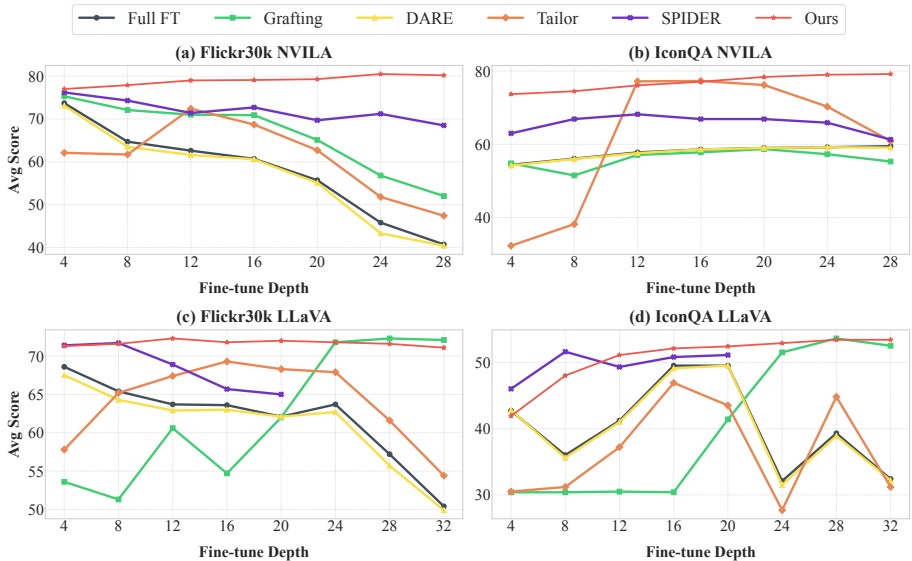

*Figure 5.* Performance comparison across fine-tuning depths on Flickr30k and IconQA. Results show the average accuracy across all tasks for an update ratio of $\rho = 0.1$ and various merging methods. The x-axis denotes the number of layers fine-tuned, counted incrementally from the final output layer toward the initial input layer.

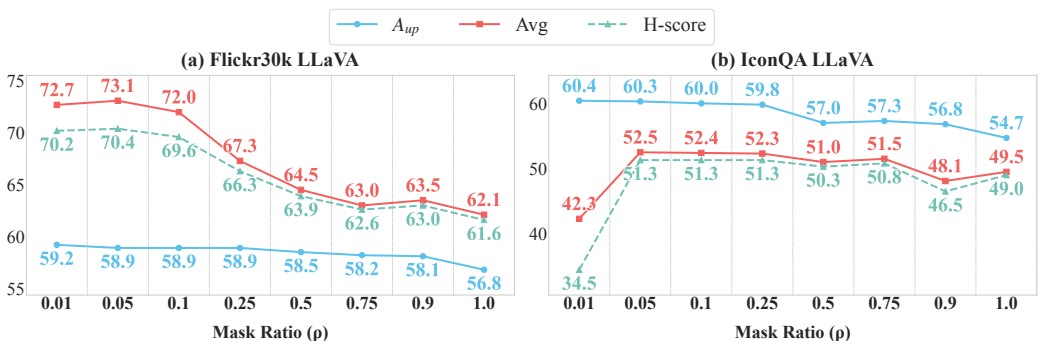

*Figure 6.* Performance comparison across various mask ratios ($\rho$) on Flickr30k and IconQA using LLaVA-1.5-7B. Results show the upstream and downstream performance.

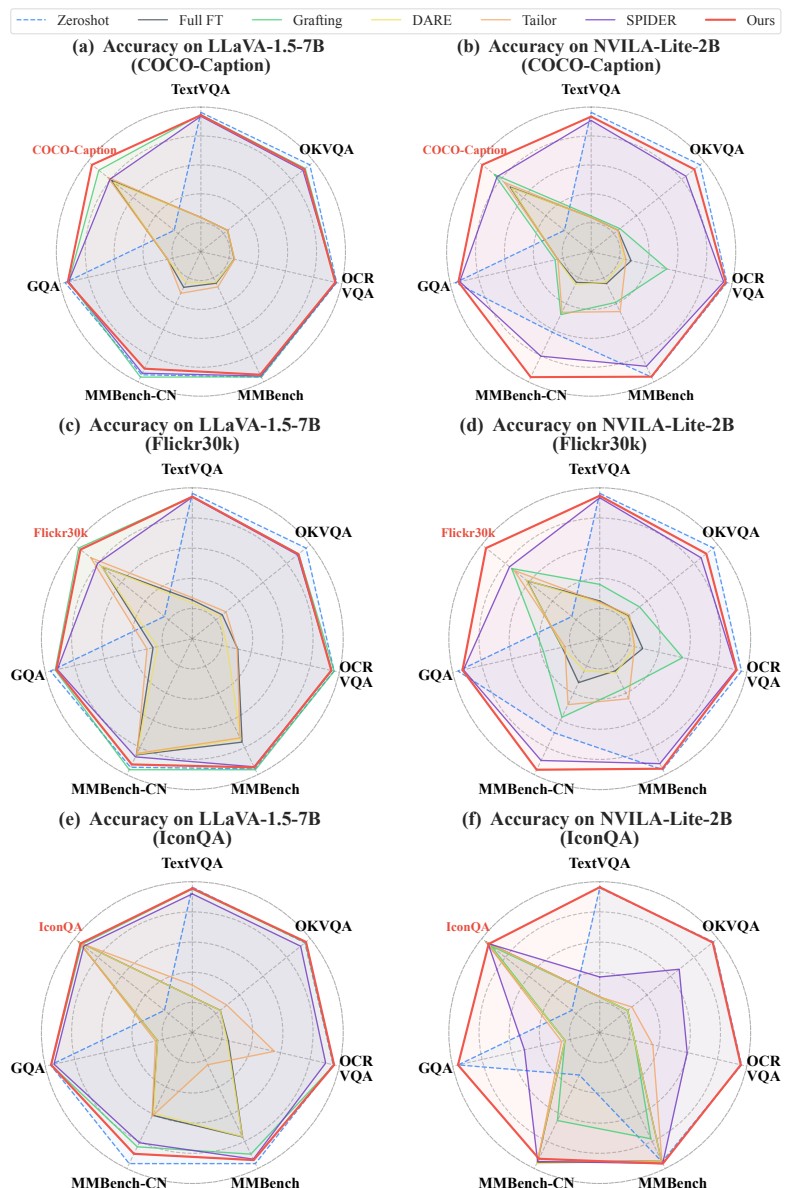

*Figure 7.* Radar chart on diverse benchmarks on LLaVA-1.5-7B and NVILA-Lite when finetuning all layers.

## F.2. Applicability on Vision Encoder

We extend Model-Dowser to jointly fine-tune the vision encoder alongside the language decoder in LLaVA-1.5-7B. Importance scores for the vision encoder follow the same Monte Carlo sampling procedure with the Hutchinson estimator as described in the main paper, except that inputs are sampled from a Gaussian distribution parameterized by ImageNet statistics. For the language decoder, importance scores are computed according to the procedure described in Section 3.2. All parameters are fine-tuned for 5 epochs with a learning rate of $2\times10^{-5}$ and $\rho = 0.1$.

As shown in Table 6, Dowser with vision encoder fine-tuning substantially outperforms Full-FT on both tasks, achieving H-scores of 79.9 on COCO and 68.9 on ImageNet-R, compared to 13.3 and 18.1 for Full-FT. This confirms that the importance-guided sparse update strategy generalizes effectively beyond the language decoder.

*Table 6.* Performance comparison when extending Model-Dowser to also fine-tune the **vision encoder** in *LLaVA-1.5-7B*, evaluated on **COCO** and **ImageNet-R**. The last 32 layers of the language decoder and the vision encoder are fine-tuned for 5 epochs (lr=$2\times10^{-5}$, $\rho$=0.1). Importance scores for the vision encoder are computed via Gaussian-sampled inputs; the language decoder follows the main paper's procedure. **Bold** and underlined entries represent the best and second-best results, respectively.

| Method | COCO | | | | | | | | | | ImageNet-R | | | | | | | | | |
|---|---|---|---|---|---|---|---|---|---|---|---|---|---|---|---|---|---|---|---|---|
| | TextVQA | OKVQA | OCRVQA | MMB | MMB(CN) | GQA | $A_{up}$ | $A_{down}$ | Avg ↑ | H-Score ↑ | TextVQA | OKVQA | OCRVQA | MMB | MMB(CN) | GQA | $A_{up}$ | $A_{down}$ | Avg ↑ | H-Score ↑ |
| Zeroshot | 58.3 | 58.0 | 65.1 | 64.6 | 58.2 | 62.0 | 61.0 | 40.3 | 50.6 | 48.5 | 58.3 | 58.0 | 65.1 | 64.6 | 58.2 | 62.0 | 61.0 | 16.3 | 38.6 | 25.7 |
| Full-FT | 0.0 | 0.5 | 2.9 | 11.2 | 28.2 | 0.0 | 7.1 | 101.8 | 54.5 | 13.3 | 11.1 | 4.6 | 6.5 | 17.2 | 18.5 | 2.5 | 10.1 | **91.2** | 50.6 | 18.1 |
| Dowser (+ Vision Enc.) | 55.8 | 54.2 | 66.3 | 64.6 | 55.8 | 59.3 | 59.3 | **122.2** | **90.8** | **79.9** | 55.1 | 46.5 | 57.8 | 64.5 | 55.3 | 53.9 | 55.5 | 90.9 | **73.2** | **68.9** |

## G. Effect of Learning Rate on Downstream and Upstream Performance

In our main experiments, we adopt the learning rates from the official fine-tuning configurations of LLaVA-1.5 (Liu et al., 2023) and NVILA (Liu et al., 2025), which are also widely used in prior works, to ensure fair and reproducible comparisons without method-specific hyperparameter tuning.

To further investigate whether the performance gap across methods is sensitive to this choice, we conduct additional experiments across three learning rates ($1 \times 10^{-4}$, $2 \times 10^{-5}$, and $5 \times 10^{-6}$) on two downstream tasks, COCO-Caption and ImageNet-R, using LLaVA-1.5-7B with the last 20 layers fine-tuned and an update ratio of $\rho = 0.1$.

As shown in Table 7, Dowser consistently achieves the highest H-scores on both tasks, while Full-FT, DARE, and Tailor exhibit substantial upstream forgetting, particularly on ImageNet-R. At high learning rates, all baselines suffer severe degradation, whereas Dowser maintains stable performance across all learning rates, consistently retaining its advantage over competing methods.

Notably, post-merging methods such as Tailor and DARE show inconsistent upstream performance as the learning rate increases, reflecting the fundamental instability of post-hoc weight fusion under large parameter shifts. In contrast, sparse fine-tuning methods demonstrate more stable behavior, with Dowser consistently outperforming SPIDER thanks to its principled, data-free importance scoring.

These results reinforce our central claim: the performance advantage of Dowser is not an artifact of any particular learning rate choice, but stems from the selective preservation of functionally important parameters, a property that post-merging methods and unconstrained fine-tuning fundamentally lack.

*Table 7.* Performance comparison on COCO-Caption and ImageNet-R downstream tasks using ***LLaVA-1.5-7B***, where only the last 20 layers of the language decoder are fine-tuned. The left block shows results for COCO-Caption, and the right block shows results for ImageNet-R. Results show the downstream performance (**Avg**, **H-score**) with an update ratio of $\rho = 0.1$. **Bold** and underlined entries represent the best and second-best results in each category.

| Method | COCO-Caption | | | | | | | | | ImageNet-R | | | | | | | | |
| --- | --- | --- | --- | --- | --- | --- | --- | --- | --- | --- | --- | --- | --- | --- | --- | --- | --- | --- |
| | TextVQA | OKVQA | OCRVQA | MMB | MMB(CN) | GQA | $A_{down}$ | Avg ↑ | H-Score ↑ | TextVQA | OKVQA | OCRVQA | MMB | MMB(CN) | GQA | $A_{down}$ | Avg ↑ | H-Score ↑ |
| Zeroshot | 58.3 | 58.0 | 65.1 | 64.6 | 58.2 | 62.0 | 40.3 | 50.7 | 48.5 | 58.3 | 58.0 | 65.1 | 64.6 | 58.2 | 62.0 | 16.3 | 38.7 | 25.8 |
| *Learning rate: $\lambda = 1 \times 10^{-4}$* | | | | | | | | | | | | | | | | | | |
| Full-FT | 39.7 | 30.4 | 50.4 | 60.4 | 52.5 | 48.0 | 97.7 | 72.3 | 63.4 | 5.3 | 3.9 | 0.7 | 10.5 | 9.4 | 1.0 | **90.1** | 47.6 | 9.7 |
| DARE | 37.9 | 29.1 | 47.2 | 57.6 | 49.6 | 47.6 | 96.9 | 70.9 | 61.3 | 5.2 | 3.8 | 0.7 | 9.6 | 10.2 | 1.1 | 90.0 | 47.6 | 9.6 |
| Tailor | 42.9 | 37.1 | 54.4 | 61.2 | 52.8 | 52.2 | 99.5 | 74.8 | 66.6 | 12.8 | 6.4 | 4.0 | 21.8 | 23.1 | 2.7 | 88.7 | 50.3 | 20.8 |
| SPIDER | 50.0 | 46.4 | 60.8 | 62.2 | 56.2 | 56.3 | **102.5** | 78.9 | 71.8 | 12.2 | 6.0 | 5.5 | 18.2 | 21.6 | 2.3 | **90.1** | 50.5 | 19.5 |
| Dowser | 53.7 | 50.2 | 62.1 | 62.7 | 56.5 | 56.4 | 102.4 | **79.7** | **73.2** | 45.5 | 26.2 | 57.9 | 58.4 | 50.0 | 42.6 | 89.4 | **68.1** | **61.4** |
| *Learning rate: $\lambda = 2 \times 10^{-5}$* | | | | | | | | | | | | | | | | | | |
| Full-FT | 52.7 | 48.8 | 63.0 | 63.9 | 58.0 | 57.5 | 100.0 | 78.7 | 72.9 | 24.3 | 10.2 | 26.2 | 48.6 | 45.0 | 20.6 | 89.7 | 59.4 | 44.0 |
| Grafting | 57.2 | 55.9 | 64.1 | 64.8 | 58.5 | 59.8 | 81.6 | 70.8 | 69.2 | 57.7 | 55.8 | 65.7 | 64.6 | 58.8 | 61.4 | 67.3 | 64.0 | 63.8 |
| DARE | 52.2 | 48.1 | 62.4 | 63.7 | 58.1 | 57.1 | 99.1 | 78.0 | 72.3 | 23.8 | 9.9 | 25.6 | 47.3 | 44.8 | 16.7 | **89.8** | 58.9 | 42.7 |
| Tailor | 55.0 | 50.9 | 64.5 | 64.8 | 58.4 | 58.8 | 106.7 | 82.7 | 75.8 | 38.1 | 18.2 | 40.1 | 61.4 | 54.8 | 36.8 | 84.3 | 62.9 | 55.7 |
| SPIDER | 56.1 | 53.2 | 64.7 | 63.9 | 57.5 | 59.8 | 103.1 | 81.1 | 75.2 | 46.8 | 27.4 | 55.6 | 63.0 | 54.8 | 43.0 | 89.3 | 68.9 | 62.8 |
| Dowser | 57.0 | 54.3 | 65.0 | 62.9 | 55.0 | 60.3 | **123.6** | **91.3** | **79.9** | 54.6 | 48.4 | 64.8 | 64.4 | 55.8 | 56.5 | 88.8 | **73.1** | **69.7** |
| *Learning rate: $\lambda = 5 \times 10^{-6}$* | | | | | | | | | | | | | | | | | | |
| Full-FT | 56.8 | 54.3 | 64.4 | 64.0 | 57.2 | 59.9 | 109.9 | 84.7 | 77.1 | 50.0 | 34.0 | 61.6 | 64.4 | 57.1 | 47.7 | **89.5** | 71.0 | 66.2 |
| DARE | 56.5 | 54.2 | 64.4 | 64.2 | 57.0 | 59.8 | 108.2 | 83.8 | 76.7 | 49.5 | 33.3 | 61.3 | 64.2 | 57.6 | 47.3 | **89.5** | 70.8 | 65.9 |
| Tailor | 57.1 | 54.5 | 64.2 | 64.6 | 58.3 | 60.3 | 115.4 | 87.6 | 78.8 | 53.6 | 43.1 | 63.2 | 64.6 | 57.6 | 53.1 | 79.4 | 67.6 | 65.6 |
| SPIDER | 57.5 | 56.4 | 64.9 | 64.1 | 57.0 | 61.5 | 125.8 | 93.0 | 81.5 | 55.0 | 47.4 | 65.0 | 64.3 | 57.2 | 55.0 | 88.5 | 72.9 | 69.6 |
| Dowser | 58.0 | 57.0 | 64.7 | 64.7 | 57.7 | 61.4 | **128.5** | **94.5** | **82.4** | 57.0 | 54.2 | 66.0 | 64.5 | 57.5 | 60.0 | 87.5 | **73.7** | **71.1** |

## H. Comparison with Random Selection

To further validate the effectiveness of our sensitivity-based importance scoring, we compare Model-Dowser with a random-selection baseline across varying update ratios ($\rho$). Tables 8 and 9 summarize the results for ImageNet-R and COCO-Caption respectively.

Model-Dowser consistently outperforms the random selection baseline across datasets and ratios. This performance gap becomes particularly evident as the update budget increases. While the difference is smaller at low sparsity ($\rho = 0.1$), the random baseline fails to effectively preserve upstream knowledge as the update ratio increases to $\rho = 0.5$. For instance, on ImageNet-R with $\rho = 0.5$, Model-Dowser achieves an H-score of 42.6 with LLaVA, significantly higher than the 33.2 obtained by random selection.

Importantly, these performance improvements are statistically significant. We conducted two-sided t-tests under the $\rho = 0.5$ condition. For NVILA-Lite-2B, the analysis produced $p$-values of 0.003 for the H-score on COCO-Caption and 0.004 for the H-score on ImageNet-R. Likewise, LLaVA-1.5-7B also shows statistically significant gains, with $p$-values of 0.08 and 0.007 on COCO-Caption and ImageNet-R, respectively. Together, these findings validate the effectiveness of the proposed importance score.

*Table 8.* Experimental result on ImageNet-R with different update ratios ($\rho$) across 28 (NVILA) and 32 (LLaVA) layers. Random selection vs ours. Numbers after $\pm$ indicate standard deviation across three random seeds.

| | Ratio ($\rho$) | TextVQA | OKVQA | OCR | MMB | MMB (CN) | GQA | $A_{up}$ | ImageNet-R | Avg ↑ | H-score ↑ |
|---|---|---|---|---|---|---|---|---|---|---|---|
| **NVILA-Lite-2B** | | | | | | | | | | | |
| Full FT | 1.0 | 12.2 | 9.3 | 2.3 | 36.5 | 17.8 | 4.9 | 13.8 | 92.3 | 53.1 | 24.0 |
| Random | 0.1 | 53.7 ±2.8 | 41.0 ±0.8 | 66.0 ±0.4 | 72.8 ±1.2 | 39.2 ±2.0 | 54.1 ±0.8 | 54.5 ±1.0 | **90.7** ±0.2 | 72.6 ±0.6 | 68.0 ±0.8 |
| Ours | | 57.9 | 43.4 | 67.7 | 75.0 | 39.4 | 56.5 | **56.6** | 90.5 | **73.6** | **69.7** |
| Random | 0.25 | 43.2 ±1.7 | 30.2 ±1.2 | 57.6 ±0.7 | 71.0 ±0.8 | 46.4 ±0.8 | 45.4 ±0.9 | 49.0 ±0.7 | 91.5 ±0.2 | 70.2 ±0.3 | 63.8 ±0.5 |
| Ours | | 50.1 | 33.1 | 63.5 | 75.0 | 47.3 | 49.3 | **53.0** | **91.7** | **72.4** | **67.2** |
| Random | 0.5 | 30.6 ±2.5 | 19.4 ±0.6 | 41.1 ±1.5 | 64.3 ±2.7 | 47.6 ±1.2 | 32.3 ±0.7 | 39.2 ±0.9 | **92.2** ±0.2 | 65.7 ±0.4 | 55.0 ±0.8 |
| Ours | | 42.3 | 25.2 | 52.6 | 71.7 | 50.7 | 42.4 | **47.5** | 92.2 | **69.8** | **62.7** |
| Random | 0.75 | 22.8 ±3.9 | 14.9 ±0.7 | 16.0 ±4.6 | 46.1 ±11.3 | 35.7 ±6.0 | 18.0 ±3.5 | 25.6 ±4.4 | **92.6** ±0.2 | 59.1 ±2.3 | 40.1 ±5.5 |
| Ours | | 30.9 | 17.1 | 41.1 | 53.1 | 38.9 | 32.0 | **35.5** | 92.5 | **64.0** | **51.3** |
| Random | 0.9 | 19.9 ±1.7 | 12.5 ±0.2 | 9.1 ±3.5 | 40.4 ±2.6 | 27.8 ±1.8 | 12.6 ±1.3 | 20.4 ±1.4 | **92.6** ±0.2 | 56.5 ±0.8 | 33.4 ±1.9 |
| Ours | | 22.0 | 13.2 | 24.9 | 41.6 | 27.5 | 20.0 | **24.9** | 92.6 | **58.7** | **39.2** |
| **LLaVA-1.5-7B** | | | | | | | | | | | |
| Full FT | 1.0 | 10.5 | 6.3 | 2.3 | 17.8 | 17.4 | 2.3 | 9.4 | 89.8 | 49.6 | 17.1 |
| Random | 0.1 | 54.2 ±0.1 | 44.3 ±0.1 | 59.3 ±0.2 | 63.0 ±0.1 | 54.9 ±0.2 | 52.3 ±0.1 | 54.6 ±0.1 | 88.5 ±0.1 | 71.6 ±0.0 | 67.6 ±0.1 |
| Ours | | 54.9 | 46.1 | 62.0 | 63.4 | 55.1 | 53.6 | **55.8** | **88.6** | **72.2** | **68.5** |
| Random | 0.25 | 45.2 ±0.1 | 28.5 ±0.2 | 41.5 ±0.7 | 53.1 ±0.3 | 49.8 ±0.1 | 41.7 ±0.1 | 43.3 ±0.2 | 89.3 ±0.1 | 66.3 ±0.1 | 58.3 ±0.2 |
| Ours | | 49.2 | 33.9 | 47.6 | 59.5 | 53.8 | 45.2 | **48.2** | **89.5** | **68.9** | **62.7** |
| Random | 0.5 | 21.4 ±0.5 | 12.3 ±0.0 | 14.7 ±0.3 | 30.3 ±0.4 | 26.7 ±0.2 | 16.8 ±0.8 | 20.4 ±0.3 | **90.0** ±0.1 | 55.2 ±0.1 | 33.2 ±0.5 |
| Ours | | 29.5 | 15.3 | 20.3 | 38.5 | 36.6 | 27.3 | **27.9** | 89.7 | **58.8** | **42.6** |
| Random | 0.75 | 13.3 ±0.1 | 7.2 ±0.1 | 7.6 ±0.2 | 21.8 ±0.1 | 18.8 ±0.7 | 3.3 ±0.3 | 12.0 ±0.2 | **89.8** ±0.2 | 50.9 ±0.1 | 21.2 ±0.3 |
| Ours | | 14.6 | 7.8 | 10.1 | 24.3 | 20.4 | 4.3 | **13.6** | 89.7 | **51.6** | **23.6** |
| Random | 0.9 | 11.0 ±0.3 | 6.5 ±0.1 | 3.1 ±0.3 | 17.9 ±0.8 | 17.9 ±0.2 | 2.4 ±0.1 | 9.8 ±0.2 | 89.7 ±0.1 | 49.8 ±0.1 | 17.6 ±0.2 |
| Ours | | 11.5 | 6.5 | 4.3 | 18.4 | 18.0 | 2.4 | **10.2** | **89.9** | **50.1** | **18.3** |

*Table 9.* Experimental result on COCO-Caption with different update ratios ($\rho$) across 28 (NVILA) and 32 (LLaVA) layers. Random selection vs ours. Numbers after $\pm$ indicate standard deviation across three random seeds.

| | Ratio ($\rho$) | TextVQA | OKVQA | OCR | MMB | MMB (CN) | GQA | $A_{up}$ | COCO-C | Avg ↑ | H-score ↑ |
|---|---|---|---|---|---|---|---|---|---|---|---|
| **NVILA-Lite-2B** | | | | | | | | | | | |
| Full FT | 1.0 | 0.2 | 0.1 | 22.5 | 25.4 | 29.3 | 0.2 | 12.9 | 98.2 | 55.6 | 22.9 |
| Random | 0.1 | 67.5 $\pm$0.2 | 47.1 $\pm$0.2 | 68.7 $\pm$0.1 | 76.9 $\pm$0.1 | 52.9 $\pm$0.4 | 60.5 $\pm$0.1 | 62.3 $\pm$0.1 | 134.8 $\pm$0.2 | 98.6 $\pm$0.1 | 85.2 $\pm$0.1 |
| Ours | | 68.0 | 47.6 | 68.8 | 77.8 | 53.8 | 60.4 | **62.7** | **135.1** | **98.9** | **85.7** |
| Random | 0.25 | 64.4 $\pm$1.0 | 43.3 $\pm$0.7 | 68.1 $\pm$0.2 | 73.1 $\pm$0.5 | 52.1 $\pm$0.2 | 58.3 $\pm$0.5 | 59.9 $\pm$0.4 | **124.3** $\pm$0.5 | 92.1 $\pm$0.2 | 80.8 $\pm$0.3 |
| Ours | | 66.6 | 45.0 | 68.7 | 75.5 | 53.2 | 59.3 | **61.4** | 123.8 | **92.6** | **82.1** |
| Random | 0.5 | 30.8 $\pm$0.6 | 26.1 $\pm$1.4 | 65.7 $\pm$0.2 | 63.9 $\pm$2.7 | 47.4 $\pm$0.8 | 35.0 $\pm$3.4 | 44.8 $\pm$1.4 | 105.2 $\pm$0.4 | 75.0 $\pm$0.7 | 62.9 $\pm$1.4 |
| Ours | | 55.0 | 35.0 | 67.2 | 71.7 | 51.5 | 51.0 | **55.2** | **106.4** | **80.8** | **72.7** |
| Random | 0.75 | 2.3 $\pm$0.4 | 4.7 $\pm$3.1 | 53.7 $\pm$5.5 | 52.1 $\pm$3.5 | 41.6 $\pm$0.2 | 8.1 $\pm$4.4 | 27.1 $\pm$2.7 | **98.6** $\pm$0.2 | 62.8 $\pm$1.3 | 42.5 $\pm$3.4 |
| Ours | | 10.4 | 14.2 | 63.9 | 62.4 | 49.1 | 20.3 | **36.7** | 98.5 | **67.6** | **53.5** |
| Random | 0.9 | 0.4 $\pm$0.2 | 1.3 $\pm$0.5 | 32.5 $\pm$1.4 | 42.8 $\pm$3.6 | 38.7 $\pm$3.2 | 2.2 $\pm$0.9 | 19.7 $\pm$0.7 | **97.8** $\pm$0.1 | 58.7 $\pm$0.3 | 32.7 $\pm$0.9 |
| Ours | | 0.9 | 2.0 | 38.9 | 49.1 | 43.6 | 3.2 | **22.9** | 97.7 | **60.3** | **37.1** |
| **LLaVA-1.5-7B** | | | | | | | | | | | |
| Full FT | 1.0 | 0.1 | 0.0 | 2.5 | 8.6 | 15.5 | 0.0 | 4.5 | 101.5 | 53.0 | 8.5 |
| Random | 0.1 | 56.5 $\pm$0.1 | 53.5 $\pm$0.2 | 64.7 $\pm$0.7 | 62.7 $\pm$0.2 | 55.5 $\pm$0.2 | 60.1 $\pm$0.4 | 58.8 $\pm$0.3 | **120.8** $\pm$0.3 | 89.8 $\pm$0.0 | 79.1 $\pm$0.2 |
| Ours | | 56.7 | 54.1 | 64.3 | 63.1 | 55.2 | 60.4 | **59.0** | 120.8 | **89.9** | **79.3** |
| Random | 0.25 | 49.0 $\pm$0.2 | 47.2 $\pm$0.0 | 63.6 $\pm$0.5 | 61.5 $\pm$0.1 | 56.5 $\pm$0.0 | 57.8 $\pm$0.5 | 55.9 $\pm$0.2 | **105.2** $\pm$0.5 | 80.6 $\pm$0.3 | 73.0 $\pm$0.3 |
| Ours | | 52.3 | 49.7 | 63.7 | 62.8 | 56.4 | 58.7 | **57.3** | 104.5 | **80.9** | **74.0** |
| Random | 0.5 | 11.9 $\pm$2.8 | 18.8 $\pm$2.1 | 32.9 $\pm$9.5 | 52.1 $\pm$1.4 | 53.5 $\pm$0.3 | 29.4 $\pm$3.4 | 33.1 $\pm$3.0 | 100.5 $\pm$0.1 | 66.8 $\pm$1.5 | 49.8$\pm$3.5 |
| Ours | | 20.0 | 26.2 | 36.6 | 54.4 | 56.4 | 41.9 | **39.2** | **100.8** | **70.0** | **56.5** |
| Random | 0.75 | 0.1 $\pm$0.0 | 1.2 $\pm$0.4 | 4.2 $\pm$0.6 | 22.4 $\pm$6.9 | 37.8 $\pm$7.9 | 0.5 $\pm$0.3 | 11.0 $\pm$2.5 | 100.5 $\pm$0.3 | 55.8$\pm$1.3 | 19.9$\pm$4.2 |
| Ours | | 0.2 | 2.2 | 6.4 | 27.2 | 43.8 | 0.9 | **13.4** | **100.9** | **57.2** | **23.7** |
| Random | 0.9 | 0.1 $\pm$0.0 | 0.2 $\pm$0.1 | 3.0 $\pm$0.2 | 11.4 $\pm$3.1 | 23.3 $\pm$6.0 | 0.0 $\pm$0.0 | 6.3 $\pm$1.5 | 100.5 $\pm$0.7 | 53.4$\pm$0.7 | 11.9$\pm$2.7 |
| Ours | | 0.1 | 0.4 | 3.2 | 12.5 | 24.0 | 0.0 | **6.7** | **101.3** | **54.0** | **12.5** |

## I. Stability of the Data-Free Estimation on the Importance Score

We provide a more detailed result that our data-free importance estimation yields a stable, robust ranking of parameter sensitivity. To evaluate the alignment between our data-free estimation and the score with real samples, we employ two metrics. First, Hamming distance (Hamming, 1950) measures the discrepancy between the binary masks of the important parameters. Specifically, consistent with our update ratio, we define the importance mask by selecting the top 10% of parameters. A lower Hamming distance indicates a higher overlap in the identified subset of critical weights. Second, Spearman's Rank Correlation (Spearman correlation) (SPEARMAN, 1904) assesses the monotonic relationship across all parameters by comparing the ranks of their importance scores. A higher coefficient indicates that our method successfully preserves the global sensitivity ranking found in the real-data setting.

Figure 8 demonstrates the stability and robustness of our approach compared to random selection. There is a significant disparity between the two methods; as the number of samples ($N$) increases, our estimated importance score closely aligns with the score rankings calculated from real data samples. This indicates that our data-free proxy effectively captures the model's true structural sensitivity.

Furthermore, quantitative validation is provided in Table 10. The results of a two-tailed paired t-test yield statistically significant $p$-values below $10^{-7}$ for both Hamming distance and Spearman correlation. These results confirm that the alignment achieved by Model-Dowser is robust and statistically distinct from random variations.

The Hamming distance and Spearman correlation both saturate as the number of Rademacher and Monte Carlo samples increases. We choose $R = 8$ and $N = 64$ for our experiments on the main paper.

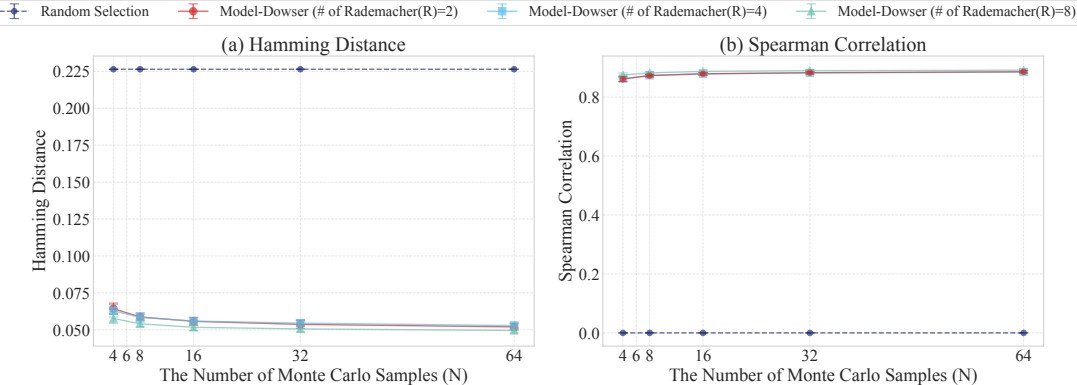

*Figure 8.* A lineplot shows hamming distance and Spearman correlation with the importance score calculated with real data samples. (a) Hamming distance (lower is better). (b) Spearman correlation (higher is better).

*Table 10.* **Statistical comparison of parameter importance** Hamming distance and Spearman rank correlation relative to the score with real-data samples. The Hamming distance is computed using binary masks that select the top 10% of parameters ($\rho = 0.1$). Results are presented as 'mean ± std' over 5 independent runs. The $p$-values are the result of a two-tailed t-test between the random baseline and ours. $R$ and $N$ denote the number of Rademacher and Monte Carlo samples, respectively.

| R | N | Hamming Distance (↓) | | | Spearman Correlation (↑) | | |
|---|---|---|---|---|---|---|---|
| | | Random Baseline | **Ours** | $p$-value | Random Baseline | **Ours** | $p$-value |
| 2 | 4 | $0.226 \pm 0.000$ | $\mathbf{0.064 \pm 0.004}$ | $1.1 \times 10^{-7}$ | $0.000 \pm 0.000$ | $\mathbf{0.861 \pm 0.008}$ | $2.9 \times 10^{-9}$ |
| | 16 | $0.226 \pm 0.000$ | $\mathbf{0.056 \pm 0.003}$ | $2.2 \times 10^{-8}$ | $0.000 \pm 0.000$ | $\mathbf{0.879 \pm 0.004}$ | $2.6 \times 10^{-10}$ |
| | 64 | $0.226 \pm 0.000$ | $\mathbf{0.052 \pm 0.002}$ | $8.6 \times 10^{-9}$ | $0.000 \pm 0.000$ | $\mathbf{0.886 \pm 0.004}$ | $1.1 \times 10^{-10}$ |
| 4 | 4 | $0.226 \pm 0.000$ | $\mathbf{0.063 \pm 0.004}$ | $1.7 \times 10^{-7}$ | $0.000 \pm 0.000$ | $\mathbf{0.862 \pm 0.009}$ | $4.1 \times 10^{-9}$ |
| | 16 | $0.226 \pm 0.000$ | $\mathbf{0.056 \pm 0.003}$ | $1.8 \times 10^{-8}$ | $0.000 \pm 0.000$ | $\mathbf{0.879 \pm 0.005}$ | $3.1 \times 10^{-10}$ |
| | 64 | $0.226 \pm 0.000$ | $\mathbf{0.053 \pm 0.002}$ | $1.1 \times 10^{-8}$ | $0.000 \pm 0.000$ | $\mathbf{0.885 \pm 0.004}$ | $1.9 \times 10^{-10}$ |
| 8 | 4 | $0.226 \pm 0.000$ | $\mathbf{0.058 \pm 0.003}$ | $5.2 \times 10^{-8}$ | $0.000 \pm 0.000$ | $\mathbf{0.875 \pm 0.007}$ | $1.6 \times 10^{-9}$ |
| | 16 | $0.226 \pm 0.000$ | $\mathbf{0.052 \pm 0.002}$ | $8.8 \times 10^{-9}$ | $0.000 \pm 0.000$ | $\mathbf{0.887 \pm 0.004}$ | $2.4 \times 10^{-10}$ |
| | 64 | $0.226 \pm 0.000$ | $\mathbf{0.050 \pm 0.002}$ | $3.0 \times 10^{-9}$ | $0.000 \pm 0.000$ | $\mathbf{0.891 \pm 0.003}$ | $7.8 \times 10^{-11}$ |

## J. Validation of Text-only Probing Strategy

We hypothesize that visual features are projected into the language embedding space and processed through the same layers as text, making the functional sensitivity of language decoder parameters similar across modalities in practice, consistent with findings (Billa, 2026) showing that decoders exhibit isotropic gradient sensitivity across modality-specific and text-aligned directions.

We verified this hypothesis empirically by comparing text-only probing against image-text probing, where synthetic image inputs are generated by sampling pixel values from a Gaussian distribution parameterized by the channel-wise mean $[0.485, 0.456, 0.406]$ and standard deviation $[0.229, 0.224, 0.225]$ of ImageNet (Russakovsky et al., 2015), following standard normalization conventions. As shown in Table 11, two strategies show almost identical rankings in terms of importance, with low Hamming distance and high Spearman correlation, suggesting that text-only probing is sufficient to capture the functional sensitivity of the language decoder parameters in MLLMs.

*Table 11.* **Effect of input modality on parameter importance** Hamming distance and Spearman rank correlation between score with image input and without image input. The Hamming distance is computed using binary masks that select the top 10% of parameters ($\rho = 0.1$). Results are presented as 'mean ± std' over 5 independent runs. $R$ and $N$ denote the number of Rademacher and Monte Carlo samples, respectively.

| R | N | Hamming Distance ($\downarrow$) | Spearman Correlation ($\uparrow$) |
|---|---|---|---|
|   | 4 | $0.092 \pm 0.002$ | $0.846 \pm 0.007$ |
| 2 | 16 | $0.083 \pm 0.003$ | $0.874 \pm 0.001$ |
|   | 64 | $0.080 \pm 0.001$ | $0.883 \pm 0.002$ |
|   | 4 | $0.091 \pm 0.003$ | $0.850 \pm 0.001$ |
| 4 | 16 | $0.083 \pm 0.002$ | $0.876 \pm 0.006$ |
|   | 64 | $0.081 \pm 0.002$ | $0.882 \pm 0.004$ |
|   | 4 | $0.092 \pm 0.002$ | $0.844 \pm 0.008$ |
| 8 | 16 | $0.084 \pm 0.001$ | $0.872 \pm 0.004$ |
|   | 64 | $0.081 \pm 0.001$ | $0.882 \pm 0.002$ |

## K. Ablation on Jacobian ($\|J\|_2$) Term

To inspect the effect of the Jacobian term on the proposed score, we provide an ablation study. We calculated Wanda (Sun et al., 2024b) style importance score, which consists of the multiplication of input activation and weight magnitude ($\|h_j^{(l-1)}\|_2 \cdot |W_{ij}^{(l)}|$). We calculate the score without the Jacobian term with synthetic input in the same way as in Section 3.2.

As shown in Table 12, our full score consistently outperforms the Wanda-style score without the Jacobian term across all update ratios in terms of H-score ($+2.2$ at $\rho = 0.1$, $+1.7$ at $\rho = 0.25$, $+0.6$ at $\rho = 0.5$). Notably, the performance gap is more pronounced at lower update ratios, suggesting that the Jacobian term is particularly critical when only a small fraction of parameters are updated, and precise identification of functionally sensitive parameters is essential.

*Table 12.* **Ablation study on the Jacobian term** ($\|J_i\|_2$). Performance comparison between the full importance score and the Wanda-style score without the Jacobian term ($|W_{ij}^{(l)}| \cdot |h_j^{(l-1)}|$) on COCO-Caption across different update ratios ($\rho$), using LLaVA-1.5-7B with the last 20 layers fine-tuned.

| | Ratio ($\rho$) | TextVQA | OKVQA | OCR | MMB | MMB (CN) | GQA | $A_{up}$ | COCO-C | Avg $\uparrow$ | H-score $\uparrow$ |
|---|---|---|---|---|---|---|---|---|---|---|---|
| Zeroshot | - | 58.3 | 58.0 | 65.1 | 64.6 | 58.2 | 62.0 | 61.0 | 40.3 | 50.7 | 48.5 |
| w/o $\|J_i\|_2$ | 0.1 | 56.0 | 53.8 | 63.6 | 63.4 | 55.2 | 58.8 | 58.4 | 116.0 | 87.2 | 77.7 |
| Ours | | 57.0 | 54.3 | 65.0 | 62.9 | 55.0 | 60.3 | **59.1** | **123.6** | **91.3** | **79.9** |
| w/o $\|J_i\|_2$ | 0.25 | 55.1 | 52.8 | 63.1 | 64.3 | 57.0 | 58.5 | 58.5 | 103.2 | 80.8 | 74.6 |
| Ours | | 56.3 | 53.7 | 64.2 | 63.1 | 56.5 | 59.6 | **58.9** | **108.2** | **83.5** | **76.3** |
| w/o $\|J_i\|_2$ | 0.5 | 54.6 | 51.6 | 62.6 | 64.1 | 58.1 | 57.7 | 58.1 | 100.7 | 79.4 | 73.7 |
| Ours | | 55.5 | 52.0 | 63.8 | 63.9 | 58.0 | 58.7 | **58.6** | **101.4** | **80.0** | **74.3** |

# L. Empirical result on Continual Learning setting

## L.1. Setup

In continual learning (CL) settings (Chen et al., 2024a; Zhao et al., 2025; Guo et al., 2025a; Chen et al., 2025; Guo et al., 2025b), catastrophic forgetting is typically studied as a byproduct of sequential task learning. In these setups, models are initialized from a pretrained backbone (often before visual instruction tuning) and then trained sequentially on a series of tasks. Although continual learning and catastrophic forgetting are often closely related, the objective of continual learning fundamentally differs from ours. CL aims to enable models to learn tasks in a strictly sequential manner, retaining knowledge from previous tasks while optimizing performance on the current task. In contrast, our work focuses on mitigating catastrophic forgetting during downstream adaptation, aiming to preserve generalization and zero-shot capabilities on unseen tasks without compromising performance on a single target downstream task. We do not assume explicit task sequences or continual task arrival; instead, we study forgetting as a consequence of fine-tuning depth and parameter updates.

To further contextualize our approach, we additionally evaluate Model-Dowser under the CL benchmark settings of (Zhao et al., 2025), but starting from visual instruction–tuned models, LLaVA-1.5-7B (Liu et al., 2024a). The benchmark consists of five downstream tasks from different domains, including: Remote Sensing (RS), Medical (Med), Autonomous Driving (AD), Science (Sci), and Financial (Fin). For baselines, we include MoELoRA (Chen et al., 2024a), a representative continual learning method, and ModelTailor (Zhu et al., 2024) as a reference post-merging approach.

**Metrics** Let $T$ be the total number of tasks and $A_{t,i}$ denote the test accuracy on task $i$ after the model has finished training on task $t$. We employ the following four standard metrics to evaluate the continual learning performance:

- **Mean Finetune Accuracy (MFT)**: This metric measures the model's plasticity, its ability to acquire new knowledge. It is calculated as the average accuracy on each task $i$ immediately after learning that specific task:

$$\text{MFT} = \frac{1}{T} \sum_{i=1}^{T} A_{i,i}. \tag{28}$$

  A higher MFT indicates that the model successfully adapts to the downstream distribution of the current task.

- **Mean Final Accuracy (MFN)**: This metric reflects the model's overall competence upon completion of the entire learning sequence. It averages the final performance on all observed tasks:

$$\text{MFN} = \frac{1}{T} \sum_{i=1}^{T} A_{T,i}. \tag{29}$$

  Unlike MFT, MFN accounts for subsequent performance degradation. A high MFN requires the model not only to learn new tasks well but also to retain performance on previous ones.

- **Mean Average Accuracy (MAA)**: This metric provides a holistic view of the model's performance stability throughout the learning process. It is defined as the average of the cumulative average accuracies recorded at each step $t$:

$$\text{MAA} = \frac{1}{T} \sum_{t=1}^{T} \left( \frac{1}{t} \sum_{i=1}^{t} A_{t,i} \right). \tag{30}$$

  MAA captures the historical trajectory of performance, rewarding models that maintain consistently high accuracy across all known tasks at every stage of training.

- **Backward Transfer (BWT)**: This is the primary metric for quantifying catastrophic forgetting. It measures the average change in accuracy for each task between the moment it was learned and the end of the sequence:

$$\text{BWT} = \frac{1}{T-1} \sum_{i=1}^{T-1} (A_{T,i} - A_{i,i}). \tag{31}$$

  A negative BWT value indicates forgetting (performance degradation on past tasks), while a value close to zero implies strong stability (retention of knowledge). Our goal is to maximize BWT (i.e., minimize the magnitude of the negative value).

*Table 13.* Performance comparison on Continual Learning (CL) settings using ***LLaVA-v1.5-7B***. Upstream tasks include TextVQA, POPE, VQAv2, MM-VET, ScienceQA-Img, MMBench (EN&CN), GQA, and SEED-Bench. The results represent the final performance after completing sequential training on 5 CL tasks.

| Method | Upstream $A_{up}$ | Downstream | | | | | | Metrics | | | | |
|---|---|---|---|---|---|---|---|---|---|---|---|---|
| | | RS | Med | AD | Sci | Fin | $A_{down}$ | MFT | MFN | MAA | BWT | H-score |
| Zeroshot | 63.9 | 32.3 | 35.5 | 15.6 | 42.3 | 62.5 | 37.6 | - | - | - | - | - |
| Full-FT | 40.2 | 60.0 | 50.1 | 20.6 | 51.0 | **91.6** | 54.7 | **70.2** | 54.7 | 66.2 | -15.6 | 46.3 |
| Tailor | 40.6 | 66.0 | 50.0 | 21.2 | 51.1 | 91.3 | 55.9 | 69.1 | 55.9 | 66.6 | -13.2 | 47.0 |
| MoELoRA | **61.2** | 73.2 | 51.5 | 35.0 | 49.6 | 90.9 | 60.0 | 67.9 | 60.0 | 64.8 | -7.8 | 60.6 |
| Dowser | **61.2** | **78.8** | **61.2** | **48.0** | **53.5** | 91.4 | **66.6** | 69.1 | **66.6** | **69.6** | **-2.5** | **63.8** |

## L.2. Experimental Result

As demonstrated in Table 13, Full-FT and Tailor exhibit significant catastrophic forgetting in the continual learning setting. They show severe forgetting of both the original pretrained knowledge (upstream Avg drops to 40.2) and downstream tasks, as evidenced by high negative Backward Transfer (BWT) scores of -15.6 and -13.2, respectively. This indicates that without explicit protection, the model overwrites previous knowledge when it adapts to a new task.

In contrast, Model-Dowser achieves remarkable performance in a continual setting. It keeps the upstream performance average of 61.2, outperforming the Zeroshot baseline and MoELoRA. Most notably, Model-Dowser achieves a BWT of -2.5, which is substantially better than MoELoRA (-7.8). This implies that our importance-score-based masking effectively isolates task-specific updates, enabling the model to learn new skills without forgetting.

Moreover, Model-Dowser does not harm learning capability. Our method achieves the highest MFN of 66.6 and an H-score of 63.8, outperforming the strongest baseline, MoELoRA (MFN: 60.0, H-score: 60.6). While Full-FT shows a slightly higher Mean Finetune Accuracy (MFT) of 70.2, its final performance collapses due to forgetting. Model-Dowser maintains a competitive MFT of 69.1 while ensuring that these gains are retained throughout the sequential training process. This confirms that our sparse fine-tuning strategy can be applied to a continual learning setting.

