# OpenReview forum: "Model-Dowser: Data-Free Importance Probing to Mitigate Catastrophic Forgetting in Multimodal Large Language Models"
_ICML.cc/2026/Conference — ICML 2026 regular_

### Official Review · Reviewer_y81P · 2026-03-07

**Soundness:** 2
**Presentation:** 3
**Significance:** 3
**Originality:** 3
**Overall Recommendation:** 4
**Confidence:** 3

**Summary:**

This paper presents a new method to mitigate catastrophic forgetting in MLLMs by calculating the importance scores of parameters in a data-free way. Specifically, they use a first order taylor series approximation to shift the output based on weight perturbation, which they use to define importance/sensitivity values of a given parameter, which extends prior work to include both the Jacobian and activation norms when calculating scores. Furthermore, as this current formulation relies on pretraining data, they generate synthetic prompts from the model, which they then use to approximate the sensitivity score. Finally, once the per-parameter sensitivity scores are calculated, they produce an update mask, so that plasticity of the model can be controlled. The combination of these methods is there method Dowser.

**Compliance With Llm Reviewing Policy:**

Affirmed.

**Final Justification:**

While I believe that the contribution is novel, I am still very hesitant about the experimental results. While I find the experimental section to include a large number of baselines, I am still very unsure about the strength of these baselines. The authors claim that full finetuning performs worse due to loss of generalization: "which also negatively impacts downstream adaptation due to generalization loss". This is counteractive to most experimental setups for catastrophic forgetting. Model Dowser is a constrained optimization method, which should underperform or match full finetuning if good hyperparameters are selected. While the authors provide a minimal sweep, I am still very hesitant about the legitimacy of each baseline's numbers. We can see that with the author's rebuttal, where they find stronger numbers for the baseline full-finetuning than what is reported in the paper. Furthermore, I am still skeptical about the validity of the following statement in the rebuttal: "This allows the model to leverage shared reasoning and multimodal structures, thereby improving performance on certain upstream tasks." This is a very strong claim, which I would be hesitant to believe due to the lack of baseline tuning.

While I am maintaining my score as I find the method to be sound in practice, the lack of robustness for hyperparameter selection is incredibly concerning, and indicates that results might be oversaturated for model dowser. If the other reviewers think that this paper should be accepted, then I will agree. However, if one of them believes that it should be rejected, I will stand behind that purely due to the appearance of poor baselines. My recommendation to the authors is to properly tune each baseline in a structured way that only looks at downstream performance for selection.

**Key Questions For Authors:**

1. How was the learning rate selected for the experiments, and were sweeps done for each baseline method, which found that all methods had relatively similar learning rates?

2. Why are MMB (CN) results for NVILA-Lite 2B with Dowser higher than zero-shot accuracy?

3. Your method is an extension of Wanda to also include the Jacobian, which measures an output sensitivity, so to what degree does this improve the performance of Dowser?

**Limitations:**

Yes

**Strengths And Weaknesses:**

Strengths
- The paper is well written and structured, with the derivation of the functional importance scoring and the accompanying method to approximate it, given zero training data, being easy to understand.
- I have loosely read the proofs, and everything appears to be valid.
- While I have some qualms with the experimental setup, the inclusion of ablations showing that Dowser's importance sampling method is more effective than pure random masking, along with a robust search over the choice of update ratios $\rho$ on several downstream tasks, strengthens the intuition of the paper.

Weaknesses
- Within the experimental setup, a single learning rate is selected for all methods, without clear justification or a sweep for each method. While this might be a non-issue, when compounded with the fact that Full-FT results have the worst downstream scores on COCO-caption, Flickr30k, and IconQA, it makes me skeptical. Dowser is clearly a constrained optimization problem, as it is masking out weight updates; however, it performs significantly better than Full-FT and, in general, all other baseline methods at downstream performance. While Dowser does appear to significantly mitigate forgetting, on MMB (CN) Dowser performs better than zeroshot model usage. In general, the experimental setup, while extensive, appears to have relatively weak baselines.
- The authors propose to use synthetic data to estimate importance scores, which is a core part of the algorithm. The included ablations in the appendix are good, but out of place within the presentation of the work, and should be discussed more in the main body.

---

> ### Author Rebuttal · Authors · 2026-03-30
>
> Thank you for your thoughtful reviews. We are glad that you found our writing and structure clear, the derivations and approximations easy to follow, and the ablation studies that strengthen the paper's overall intuition. Below, we address each of your concerns in detail.
> >### W1. robustness to learning rate selection and understanding the seemingly strong performance
> >
> We thank the reviewer for this concern regarding experimental fairness and baseline strength. We address this from two aspects: **(i) robustness to learning rate selection (please see Q1)** and **(ii) understanding the seemingly strong performance, including improvements over zero-shot (please see Q2)**.
> >### W2. Location of the ablation study on sensitivity of synthetic probing
> >
> We thank the reviewer for this valuable feedback. We agree that the synthetic probing validation is a core part of our method and should be discussed more in the main body. In a revised version, we will move the key results from Appendix G to the main text. For the reviewer's convenience, we summarize the key findings of Appendix G: our data-free estimation achieves strong alignment with real data (Spearman correlation above 0.88, Hamming distance below 0.065) and remains robust even with only N=4 Monte Carlo samples, as detailed in our response on Q1 of Reviewer 9VTv.
> >### Q1. Learning rate selection
> >
> We thank the reviewer for raising this important concern. In our main experiments, we adopt the learning rate from the official fine-tuning configurations of LLaVA-1.5 [1] and NVILA [2], which is also widely used in prior works to ensure fair and reproducible comparisons rather than doing hyperparameter tuning for each method.
>
> To further investigate the lower downstream performance of Full-FT, we conducted additional experiments across multiple learning rates (1e-4, 2e-5, 5e-6). As shown in the results, Full-FT consistently underperforms our method across all learning rates, both in upstream retention and H-score. This indicates that the performance gap is not due to suboptimal hyperparameter selection but rather to catastrophic forgetting during unconstrained fine-tuning, which also negatively impacts downstream adaptation due to generalization loss.
>
> These results reinforce our key claim: preserving functionally important parameters is critical, and simply adjusting optimization hyperparameters cannot resolve forgetting.
>
> **LR sweep on COCO (5 epoch, 20 layers):**
>
> |Method|$A_{up}$|$A_{down}$|Avg|H-score|
> |-|-|-|-|-|
> |lr 1e-4|||||
> |Full-FT|46.9|97.7|72.3|63.4|
> |Dowser|**56.9**|**102.4**|**79.7**|**73.2**|
> |lr 2e-5|||||
> |Full-FT|57.3|100.0|78.7|72.9|
> |Dowser|**59.1**|**123.6**|**91.3**|**79.9**|
> |lr 5e-6|||||
> |Full-FT|59.4|109.9|84.7|77.1|
> |Dowser|**60.6**|**128.5**|**94.5**|**82.4**|
> >### Q2. Strong performance over zero-shot baseline
> >
> Regarding the observation that our method occasionally surpasses zero-shot performance (e.g., on MMB-CN), this is possibly a result of positive transfer rather than overfitting. Prior work (e.g., LLaVA, Sec. 4.3) [1] has shown that models can achieve strong performance on MMBench-CN even without explicit Chinese visual instruction tuning data, indicating that cross-modal and cross-lingual transfer can arise in MLLMs.
>
> In our case, this effect is further enabled by our method: by preserving functionally important parameters, the model retains its pretrained capabilities while benefiting from improved alignment introduced during fine-tuning. This allows the model to leverage shared reasoning and multimodal structures, thereby improving performance on certain upstream tasks. In contrast, Full-FT disrupts these parameters, which limits such transfer and results in degradation.
>
> >### Q3. Effect of Jacobian term
> >
> We respectfully clarify that Model-Dowser is not an extension of Wanda, but a theoretically derived framework with fundamental differences in its objective and Jacobian formulation, as detailed in our response to W1 of Reviewer 8bTm. Here, we focus on the empirical contribution of the Jacobian term. We conduct an additional ablation study on $\|J_i\|_2$ scoring with LLaVA-1.5-7B (last 20 layers) fine-tuned on COCO across multiple update ratios:
>
> **Ablation on $\|\|J_i\|\|_2$ term with COCO:**
>
> |Method|$A_{up}$|$A_{down}$|Avg|H-score|
> |-|-|-|-|-|
> |w/o $\| \|J_i\|\|_2$ ($\rho=0.1)$|58.4|116.0|87.2|77.7|
> |Ours $(\rho=0.1)$|59.1|123.6|**91.3**|**79.9**|
> |w/o $\| \|J_i\| \|_2$ ($\rho=0.25)$|58.5|103.2|80.8|74.6|
> |Ours$(\rho=0.25)$|58.9|108.2|**83.5**|**76.3**|
> |w/o $\| \|J_i\| \|_2$ ($\rho=0.5)$|58.1|100.7|79.4|73.7|
> |Ours $(\rho=0.5)$|58.6|101.4|**80.0**|**74.3**|
>
> As shown in the table, ours shows better results in both downstream performance and H-score, confirming that the $||J_i||_2$ term captures output sensitivity beyond what weight magnitudes and activations alone provide.
>
> >### References
> >
> [1] Improved Baselines with Visual Instruction Tuning. CVPR2024
>
> [2] NVILA: Efficient Frontier Visual Language Models. CVPR2025

---

> > ### Author Rebuttal · Reviewer_y81P · 2026-04-02
> >
> > Most of my questions have been answered. I maintain my score.

---

> > > ### Author Response · Authors · 2026-04-07
> > >
> > > Thanks for your constructive feedback.
> > >
> > > We are glad to provide additional clarification on our rebuttal:
> > >
> > > > Strong performance over Full-FT baseline
> > > >
> > >
> > > We respectfully clarify that this phenomenon is not counteractive to the experimental setting for catastrophic forgetting in MLLMs. This exact tendency is consistently observed in recent literature. For instance, as shown in Tables 4 and 5 of Huang et al. [3] (ICML 2025), the SPIDER, a sparse fine-tuning approach, also significantly outperforms the Full-FT baselines. Also, as shown in the tables below, several strong baseline methods, including SPIDER [3] and Tailor [4], can outperform Full-FT on downstream tasks.  This supports our claim that Full-FT performs worse on downstream tasks due to severe generalization loss and structural collapse of the pre-trained representations.
> > >
> > > > Baseline Tuning.
> > > >
> > >
> > > To address concerns about the lack of baseline tuning, we conducted additional experiments using LLaVA-1.5 7B across multiple learning rates (1e-4, 2e-5, and 5e-6) for all methods on COCO-captioning and ImageNet-R. Model-Dowser outperforms all baselines across all learning rates in both Avg. and H-score; notably, even the optimal baseline settings fall short of our performance (e.g., 94.5 for Dowser vs. 93.0 for SPIDER at lr 5e-6 on COCO Avg.). These results confirm that the performance gap does not stem from under-optimized baselines but rather reflects the consistent advantage of explicitly preserving functionally important parameters during adaptation.
> > >
> > > **Result across Learning Rate on COCO (5 epochs, fine-tune 20 layers, $\rho=0.1$):**
> > >
> > > |  | lr 1e-4 |  |  |  |  | lr 2e-5 |  |  |  |  | lr 5e-6 |  |  |  |
> > > | --- | --- | --- | --- | --- | --- | --- | --- | --- | --- | --- | --- | --- | --- | --- |
> > > | Method | $A_{up}$ | $A_{down}$ | Avg | H-score |  | $A_{up}$ | $A_{down}$ | Avg | H-score |  | $A_{up}$ | $A_{down}$ | Avg | H-score |
> > > | Full-FT | 46.9 | 97.7 | 72.3 | 63.4 |  | 57.3 | 100.0 | 78.7 | 72.9 |  | 59.4 | 109.9 | 84.7 | 77.1 |
> > > | Grafting  | 48.1 | 99.8 | 74.0 | 64.9 |  | 60.0 | 81.6 | 70.8 | 69.2 |  | 60.7 | 77.3 | 69.0 | 68.0 |
> > > | DARE  | 44.8 | 96.9 | 70.9 | 61.3 |  | 56.9 | 99.1 | 78.0 | 72.3 |  | 59.3 | 108.2 | 83.8 | 76.7 |
> > > | Tailor  | 50.1 | 99.5 | 74.8 | 66.6 |  | 58.7 | 106.7 | *82.7* | *75.8* |  | 59.8 | 115.4 | 87.6 | 78.8 |
> > > | SPIDER  | 55.3 | 102.5 | *78.9* | *71.8* |  | 59.2 | 103.1 | 81.1 | 75.2 |  | 60.2 | 125.8 | *93.0* | *81.5* |
> > > | Dowser (Ours) | 56.9 | 102.4 | **79.7** | **73.2** |  | 59.1 | 123.6 | **91.3** | **79.9** |  | 60.6 | 128.5 | **94.5** | **82.4** |
> > >
> > > **Result across Learning Rate on ImageNet-R (5 epochs, fine-tune 20 layers, $\rho=0.1$):**
> > >
> > > |  | lr 1e-4 |  |  |  |  | lr 2e-5 |  |  |  |  | lr 5e-6 |  |  |  |
> > > | --- | --- | --- | --- | --- | --- | --- | --- | --- | --- | --- | --- | --- | --- | --- |
> > > | Method | $A_{up}$ | $A_{down}$ | Avg | H-score |  | $A_{up}$ | $A_{down}$ | Avg | H-score |  | $A_{up}$ | $A_{down}$ | Avg | H-score |
> > > | Full-FT | 5.1 | 90.1 | 47.6 | 9.7 |  | 29.1 | 89.7 | 59.4 | 44.0 |  | 52.5 | 89.5 | 71.0 | 66.2 |
> > > | Grafting  | 8.4 | 89.2 | 48.8 | 15.3 |  | 60.7 | 67.3 | 64.0 | *63.8* |  | 53.5 | 89.2 | 71.4 | 66.9 |
> > > | DARE  | 5.1 | 90.0 | 47.6 | 9.6 |  | 28.0 | 89.8 | 58.9 | 42.7 |  | 52.2 | 89.5 | 70.8 | 65.9 |
> > > | Tailor  | 11.8 | 88.7 | 50.3 | *20.8* |  | 41.6 | 84.3 | 62.9 | 55.7 |  | 55.8 | 79.4 | 67.6 | 65.6 |
> > > | SPIDER  | 11.0 | 90.1 | *50.5* | 19.5 |  | 48.4 | 89.3 | *68.9* | 62.8 |  | 57.3 | 88.5 | *72.9* | *69.6* |
> > > | Dowser (Ours) | 46.7 | 89.4 | **68.1** | **61.4** |  | 57.4 | 88.8 | **73.1** | **69.7** |  | 59.8 | 87.5 | **73.7** | **71.1** |
> > >
> > > We will include this analysis in the final manuscript.
> > >
> > > > **References**
> > > >
> > >
> > > [3] Learn from Downstream and Be Yourself in Multimodal Large Language Model Fine-Tuning. ICML 2025
> > >
> > > [4] Model Tailor: Mitigating Catastrophic Forgetting in Multi-modal Large Language Models. ICML2024

---

### Official Review · Reviewer_8bTm · 2026-03-12

**Soundness:** 2
**Presentation:** 3
**Significance:** 2
**Originality:** 2
**Overall Recommendation:** 4
**Confidence:** 3

**Summary:**

This article proposes a sparse fine-tuning method Model-Dowser for MLLMs. Specifically, Model-Dowser measures an importance metric for each parameter by jointly considering weight magnitudes, input activations, and output sensitivities. During fine-tuning, Model-Dowser preserves the highly important parameters. Experiments on LLaVA and NVILA demonstrate the effectiveness of Model-Dowser on mitigating catastrophic forgetting.

**Compliance With Llm Reviewing Policy:**

Affirmed.

**Final Justification:**

The main results, i.e., Table 1 and 2 in the article, show that Model-Dowser leads to a significant improvements over Full-FT and the comparing baselines. The authors managed to provide experimental comparisons with LoRASculpt, and it outperforms all initially compared baselines in the article. I remain skeptical about whether there are more good-performing baselines, and whether the existing baselines are fully optimized on the chosen benchmarks. Therefore, I maintain my score.

Edit 1: The authors have provided follow-up explanations for the baseline performance. I am not quite sure about their results, but considering the efforts they put in, I have raised my score and lowered my confidence.

**Key Questions For Authors:**

See weaknesses.

**Limitations:**

Yes

**Strengths And Weaknesses:**

**Strengths**:

1. **Simplicity.** The importance indicators in Model-Dowser is simple and computationally light to calculate. During training, Model-Dowser computes a static importance mask before downstream adaptation and does not require access to pretraining data, making the method easy to use in practice.
2. **Broad experimental results.** Extensive comparisons on various ablations on different design choices are conducted to demonstrate the effectiveness of the current design.
3. **Writing.** The overall intuition is easy to follow and the big idea is clear. The theory is well-presented and simple to understand.

**Weaknesses**:

1. **Novelty.** The proposed importance score is a direct combination of previously used importance signals: Wanda (arXiv:2306.11695) uses weight$\times$activation for LLM pruning, while many gradient-based pruning methods such as LLMPruner (arXiv:2305.11627) argue that first-order sensitivity reflects importance. Model-Dowser feels more like a direct combination of existing importance heuristics than proposing a new criterion.
2. **Data-free importance estimation design.** The method estimates importance using synthetic prompts produced from random tokens. There is no comparison to stronger alternatives such as using public generic text or a small calibration set gathered from MLLM fine-tuning datasets (such as from LLaVA training set). More importantly, how does text-only probing identify parameters that matter for multi-modal training with MLLMs?
3. **Evaluation Concerns.** Model-Dowser seems to improve by a large margin compared to the listed baselines. However, recent works on forgetting+MLLMs, such as [1], is missing.

[1] LoRASculpt: Sculpting LoRA for Harmonizing General and Specialized Knowledge in Multimodal Large Language Models (CVPR 2025)

---

> ### Author Rebuttal · Authors · 2026-03-30
>
> Thank you for your thoughtful reviews. We appreciate your recognition of our method's simplicity and practicality, the breadth of our experiments, and the clarity of our writing. Below, we address each of your concerns in detail.
> > ### W1. Novelty
> >
> We respectfully disagree with the reviewer's comments. We highlight two key differences from prior works:
>
> **The objective of what a score quantifies.** Wanda is a bottom-up pruning heuristic, and LLM-Pruner minimizes task-dependent loss variation ($\Delta L$), requiring calibration data to prune a model. In contrast, Model-Dowser bounds the functional shift ($||\Delta f||_2$) via a first-order Taylor approximation
> without any calibration data, an objective distinct from prior work, yielding our score as a mathematical consequence rather than a design choice.
>
> **Contribution on Jacobian term.** The non-trivial contribution is not the use of a Jacobian term itself, but the identification that $J_i=\partial f/\partial z_i$ is task-free and estimable without any calibration data. Prior gradient-based methods, which utilize $\partial L / \partial W$, assumed that meaningful sensitivity estimates require a task-specific loss with a calibration dataset. Our framework does not require any dataset, as $J_i$ captures the model's functional sensitivity independently of downstream tasks and can be estimated via synthetic probing. This distinguishes our method from prior work.
>
> An ablation comparing our score against Wanda-style scoring (w/o $||J_i||_2$) confirms consistent improvements in H-score and downstream performance (please refer to Q3 of Reviewer y81P), validating this as a meaningful contribution beyond a combinining existing methods.
> > ### W2. Validation of Data-free importance estimation
> >
> **Comparison with the calibration set.** Please refer to response on Q1 of Reviewer 9VTv.
>
> **Validity of text-only probing.** We hypothesize that visual features are projected into the language embedding space and processed through the same layers as text, making the functional sensitivity of language decoder parameters similar across modalities in practice, consistent with findings [1] showing that decoders exhibit isotropic gradient sensitivity across modality-specific and text-aligned directions. We verified this empirically by comparing text-only against image-text probing (Gaussian-sampled image tokens with ImageNet statistics):
>
> **Comparison of importance score (image-text vs. text-only):**
> |R|N|Hamming Dist. (↓)|Spearman Corr. (↑)|
> |-|-|-|-|
> |2|4|0.092 ± 0.002|0.846 ± 0.007|
> |2|64|0.080 ± 0.001|0.883 ± 0.002|
> |8|4|0.092 ± 0.002|0.844 ± 0.008|
> |8|64|0.081 ± 0.001|0.882 ± 0.002|
>
> Text-only probing maintains a Spearman correlation above 0.84 across all settings, indicating that the ranking is preserved regardless of the image input. To further validate this, we fine-tune the LLaVa-1.5-7B model with two strategies (with image and without image) (5 epochs, lr 2e-5, last 20 layers, ρ=0.1):
>
> **COCO:**
> |Method|$A_{up}$|$A_{down}$|Avg|H-score|
> |-|-|-|-|-|
> |Image-text probing|59.1|123.5|91.3|79.9|
> |Text-only probing|59.1|123.6|91.3|80.0|
>
> **ImageNet-R:**
> |Method|$A_{up}$|$A_{down}$|Avg|H-score|
> |-|-|-|-|-|
> |Image-text probing|56.9|88.8|72.8|69.3|
> |Text-only probing|57.4|88.8|73.1|69.7|
>
> The two strategies yield almost identical results, confirming that text-only probing identifies functionally equivalent importance masks while avoiding the additional complexity of handling image tokens.
>
> > ### W3. Evaluation Concerns
> >
> We thank the reviewer for pointing out LoRASculpt. Note that the two methods operate under different design spaces: LoRASculpt applies *LoRA-based parameter-efficient fine-tuning*, whereas our method targets *full-parameter fine-tuning* with importance-guided sparse updates, which inherently differ in forgetting dynamics. Nevertheless, we include LoRASculpt as a reference point, using results from the LoRASculpt paper for COCO and reproduced results for ImageNet-R (32 layers, ρ=0.1, excluding MMBench for fairness). Our method outperforms LoRASculpt: H-score of 79.2 vs. 75.4 on COCO, and 70.7 vs. 66.7 on ImageNet-R, demonstrating stronger control over the adaptation-forgetting trade-off.
>
> **COCO:**
> |Method|$A_{up}$|$A_{down}$|Avg|H-score|
> |-|-|-|-|-|
> |Zeroshot|60.8|40.3|50.6|48.5|
> |Full-FT|0.7|101.5|51.1|1.3|
> |LoRA (rank=16)|52.6|112.2|82.4|71.6|
> |LoRASculpt (rank=16)|54.7|121.3|*88.0*|*75.4*|
> |LoRA (rank=32)|49.9|110.3|80.1|68.7|
> |LoRASculpt (rank=32)|54.3|120.4|87.3|74.8|
> |Dowser|58.9|120.8|**89.8**|**79.2**|
>
> **ImageNet-R:**
> |Method|$A_{up}$|$A_{down}$|Avg|H-score|
> |-|-|-|-|-|
> |Zero-Shot|60.8|16.3|38.6|25.7|
> |Full-FT|5.4|89.8|47.6|10.1|
> |LoRA (rank=16)|38.6|89.8|64.2|54.0|
> |LoRASculpt (rank=16)|53.5|88.6|*71.1*|*66.7*|
> |LoRA (rank=32)|16.3|89.9|53.1|27.5|
> |LoRASculpt (rank=32)|52.3|88.8|70.6|65.9|
> |Dowser|58.9|88.6|**73.7**|**70.7**|
> >### References
> >
> [1] Billa, J. Modality Collapse as Mismatched Decoding: Information-Theoretic Limits of Multimodal LLMs. 2026.

---

> > ### Author Rebuttal · Reviewer_8bTm · 2026-04-05
> >
> > Thanks for the authors' detailed response. Most of my concerns are resolved. There are some following questions upon reading the response:
> >
> > 1. The Full-FT baseline, and other selected comparison methods, don't seem as fully optimized on the chosen evaluation benchmarks. I remain skeptical about the significance of the improvements by Model-Dowser.

---

> > > ### Author Response · Authors · 2026-04-07
> > >
> > > We are glad that our rebuttal has addressed most of the concerns.
> > >
> > > Regarding the additional question:
> > >
> > > > The Full-FT baseline, and other selected comparison methods, don't seem as fully optimized on the chosen evaluation benchmarks. I remain skeptical about the significance of the improvements by Model-Dowser.
> > > >
> > >
> > > We would like to clarify that all baselines and our method in the main experiments are implemented using the official fine-tuning configurations for LLaVA-1.5 [2] and NVILA [3], consistent with the settings reported in the baseline papers [4-7], to ensure fair and reproducible comparisons rather than per-method hyperparameter tuning.
> > >
> > > To further validate that the observed performance differences are not due to suboptimal tuning, we conducted additional experiments using LLaVA-1.5 7B across multiple learning rates (1e-4, 2e-5, and 5e-6) for all methods on COCO-captioning and ImageNet-R. Model-Dowser outperforms all baselines across all learning rates in both Avg. and H-score; notably, even the optimal baseline settings fall short of our performance (e.g., 94.5 for Dowser vs. 93.0 for SPIDER at lr 5e-6 on COCO Avg.). These results confirm that the performance gap does not stem from under-optimized baselines but rather reflects the consistent advantage of explicitly preserving functionally important parameters during adaptation.
> > >
> > > **Result across Learning Rate on COCO (5 epochs, fine-tune 20 layers, $\rho=0.1$):**
> > >
> > > |  | lr 1e-4 |  |  |  |  | lr 2e-5 |  |  |  |  | lr 5e-6 |  |  |  |
> > > | --- | --- | --- | --- | --- | --- | --- | --- | --- | --- | --- | --- | --- | --- | --- |
> > > | Method | $A_{up}$ | $A_{down}$ | Avg | H-score |  | $A_{up}$ | $A_{down}$ | Avg | H-score |  | $A_{up}$ | $A_{down}$ | Avg | H-score |
> > > | Full-FT | 46.9 | 97.7 | 72.3 | 63.4 |  | 57.3 | 100.0 | 78.7 | 72.9 |  | 59.4 | 109.9 | 84.7 | 77.1 |
> > > | Grafting [4] | 48.1 | 99.8 | 74.0 | 64.9 |  | 60.0 | 81.6 | 70.8 | 69.2 |  | 60.7 | 77.3 | 69.0 | 68.0 |
> > > | DARE [5] | 44.8 | 96.9 | 70.9 | 61.3 |  | 56.9 | 99.1 | 78.0 | 72.3 |  | 59.3 | 108.2 | 83.8 | 76.7 |
> > > | Tailor [6] | 50.1 | 99.5 | 74.8 | 66.6 |  | 58.7 | 106.7 | *82.7* | *75.8* |  | 59.8 | 115.4 | 87.6 | 78.8 |
> > > | SPIDER [7] | 55.3 | 102.5 | *78.9* | *71.8* |  | 59.2 | 103.1 | 81.1 | 75.2 |  | 60.2 | 125.8 | *93.0* | *81.5* |
> > > | Dowser (Ours) | 56.9 | 102.4 | **79.7** | **73.2** |  | 59.1 | 123.6 | **91.3** | **79.9** |  | 60.6 | 128.5 | **94.5** | **82.4** |
> > >
> > > **Result across Learning Rate on ImageNet-R (5 epochs, fine-tune 20 layers, $\rho=0.1$):**
> > >
> > > |  | lr 1e-4 |  |  |  |  | lr 2e-5 |  |  |  |  | lr 5e-6 |  |  |  |
> > > | --- | --- | --- | --- | --- | --- | --- | --- | --- | --- | --- | --- | --- | --- | --- |
> > > | Method | $A_{up}$ | $A_{down}$ | Avg | H-score |  | $A_{up}$ | $A_{down}$ | Avg | H-score |  | $A_{up}$ | $A_{down}$ | Avg | H-score |
> > > | Full-FT | 5.1 | 90.1 | 47.6 | 9.7 |  | 29.1 | 89.7 | 59.4 | 44.0 |  | 52.5 | 89.5 | 71.0 | 66.2 |
> > > | Grafting [4] | 8.4 | 89.2 | 48.8 | 15.3 |  | 60.7 | 67.3 | 64.0 | *63.8* |  | 53.5 | 89.2 | 71.4 | 66.9 |
> > > | DARE [5] | 5.1 | 90.0 | 47.6 | 9.6 |  | 28.0 | 89.8 | 58.9 | 42.7 |  | 52.2 | 89.5 | 70.8 | 65.9 |
> > > | Tailor [6] | 11.8 | 88.7 | 50.3 | *20.8* |  | 41.6 | 84.3 | 62.9 | 55.7 |  | 55.8 | 79.4 | 67.6 | 65.6 |
> > > | SPIDER [7] | 11.0 | 90.1 | *50.5* | 19.5 |  | 48.4 | 89.3 | *68.9* | 62.8 |  | 57.3 | 88.5 | *72.9* | *69.6* |
> > > | Dowser (Ours) | 46.7 | 89.4 | **68.1** | **61.4** |  | 57.4 | 88.8 | **73.1** | **69.7** |  | 59.8 | 87.5 | **73.7** | **71.1** |
> > >
> > > We will include this analysis in the final manuscript.
> > >
> > > > **References**
> > > >
> > >
> > > [2] Improved Baselines with Visual Instruction Tuning. CVPR2024
> > >
> > > [3] NVILA: Efficient Frontier Visual Language Models. CVPR2025
> > >
> > > [4] Task-Specific Skill Localization in Fine-tuned Language Models. ICML2023
> > >
> > > [5] Language Models are Super Mario: Absorbing Abilities from Homologous Models as a Free Lunch. ICML2024
> > >
> > > [6] Model Tailor: Mitigating Catastrophic Forgetting in Multi-modal Large Language Models. ICML2024
> > >
> > > [7] Learn from Downstream and Be Yourself in Multimodal Large Language Model Fine-Tuning. ICML 2025

---

### Official Review · Reviewer_9VTv · 2026-03-12

**Soundness:** 4
**Presentation:** 4
**Significance:** 3
**Originality:** 3
**Overall Recommendation:** 4
**Confidence:** 2

**Summary:**

This paper studies catastrophic forgetting in multimodal large language models during downstream fine-tuning. The authors argue that existing mitigation methods either become unreliable when deeper decoder layers are tuned or scale poorly in memory and computation. To address this, they propose Model-Dowser, a data-free sparse fine-tuning method that estimates a parameter importance score by combining output sensitivity, weight magnitude, and input activation, using synthetic probing plus a Hutchinson-style Jacobian approximation; then it freezes high-importance weights and updates only the less important ones during adaptation.

**Compliance With Llm Reviewing Policy:**

Affirmed.

**Key Questions For Authors:**

1. Your method works well with synthetic probing, but how sensitive is it to the quality or diversity of those synthetic prompts? If the probes are a bit weak or off-distribution, does the mask still hold up?

2. A lot of the motivation is about deeper fine-tuning hurting early decoder layers. Do you think the gain is mainly from protecting those early layers better, or is the importance scoring helping in a more general way across all depths?

3. The experiments mainly fine-tune the language decoder while keeping the other components frozen. If you also adapted the vision side, do you expect Model-Dowser to still work as cleanly, or would the importance estimation become much harder?

**Limitations:**

yes

**Strengths And Weaknesses:**

The paper has a clear motivation and a fairly coherent method design. It identifies an important practical issue in MLLM adaptation, namely that forgetting becomes much worse when fine-tuning goes deeper into the language decoder, and it responds with a method that is easy to understand at a high level: estimate which parameters matter most for preserving pretrained behavior, then avoid changing them.

The main weakness is that the paper’s core idea still feels somewhat close to existing sparse fine-tuning and parameter-importance methods, so the novelty may come across more as a better scoring rule and probing recipe than as a fundamentally new continual-learning framework.

---

> ### Author Rebuttal · Authors · 2026-03-30
>
> Thank you for your thoughtful reviews. We appreciate your positive comments on our clear motivation, coherent method design, and practical solution to the deeper fine-tuning forgetting problem. Below, we address each of your concerns in detail.
>
> > ### W1. Novelty over existing sparse fine-tuning methods
> >
>
> We directly address the reviewer's characterization of "a better scoring rule and probing recipe rather than a fundamentally new continual-learning framework." We agree that Model-Dowser is not a new continual-learning paradigm; our contribution is a diagnostic finding (existing methods fail at deeper fine-tuning) paired with a principled, scalable solution. However, we clarify that our score is derived top-down from bounding $\|\|\Delta f\|\|_2$, not a combination of the prior scoring method. The data-free probing is likewise not merely convenient but enabled by the insight that $J_i$ is task-independent. (Please refer to our response W1 of Reviewer 8bTm and Q3 of Reviewer y81P for further details.)
>
> > ### Q1. Sensitivity of synthetic probing
> >
>
> Thank you for the insightful question. To validate this, we compared our synthetic probing against real image-text pairs from downstream datasets (TextVQA, OKVQA, OCRVQA, MMBench, MMBench-CN, GQA) on LLaVA 1.5 7B, using Hamming Distance (bottom 10% parameters) and Spearman Rank Correlation (all parameters), averaged over 5 independent runs. Note that the real-sample reference scores are computed from N=64 Monte Carlo samples and R=8 Rademacher vectors. Our Monte Carlo averaging across N synthetic samples serves as a variance reduction mechanism, ensuring that the identified important parameters are robust across diverse activation patterns. As shown in the table, even with N=4 samples, the importance ranking remains stable with high Spearman correlation, suggesting that the mask does not rely on the quality or diversity of individual probes. The consistently low Hamming distance and high Spearman correlation across all settings confirm that our data-free estimation robustly captures the model's structural sensitivity. (More details in Appendix G)
>
> |R|N|Hamming Dist. (↓)|Spearman Corr. (↑)|
> |-|-|-|-|
> |2|4|0.064 ± 0.004|0.861 ± 0.008|
> |2|64|0.052 ± 0.002|0.886 ± 0.004|
> |8|4|0.058 ± 0.003|0.875 ± 0.007|
> |8|64|0.050 ± 0.002|0.891 ± 0.003|
>
> > ### Q2. Impact of scoring on deeper fine-tuning
> >
>
> We clarify that Model-Dowser applies the update ratio independently within each fine-tuned layer rather than globally across all parameters, since importance-score distributions vary across layers due to distinct activation and weight statistics. As a result, our method does not simply protect a subset of early layers; instead, it performs layer-wise sparse adaptation, allowing every selected layer, including early layers, to contribute to downstream learning while preserving the most functionally important parameters within that layer. We will update the experiment settings in the final version for clarification.
>
> >### Q3. Applicability on the vision encoder
> >
>
> Thank you for pointing this out. To address this question, we extend Model-Dowser to also fine-tune the vision encoder in LLaVA-1.5-7B. For the vision encoder, importance scores are computed using Gaussian-sampled inputs, while the language decoder follows the same procedure as described in the main paper (Monte Carlo sampling with Hutchinson estimator). All parameters are fine-tuned for 5 epochs with a learning rate of 2e-5.
>
> **COCO (32 layers, ρ=0.1):**
>
> |Method|$A_{up}$|$A_{down}$|Avg|H-score|
> |-|-|-|-|-|
> |Zeroshot|61.0|40.3|50.7|48.5|
> |Full-FT (Vision tuned)|7.1|101.8|54.5|13.3|
> |Dowser (Vision tuned)|59.3|122.2|**90.8**|**79.9**|
>
> **ImageNet-R (32 layers, ρ=0.1):**
>
> |Method|$A_{up}$|$A_{down}$|Avg|H-score|
> |-|-|-|-|-|
> |Zeroshot|61.0|16.3|38.7|25.8|
> |Full-FT (Vision tuned)|10.1|91.2|50.6|18.1|
> |Dowser (Vision tuned)|55.5|90.9|**73.2**|**68.9**|
>
> As shown in the Tables, Model-Dowser effectively preserves upstream performance while adapting to downstream tasks even when the vision encoder is also fine-tuned, demonstrating that our importance estimation remains reliable beyond the language decoder. The consistent results across both COCO captioning and ImageNet-R suggest that the method generalizes to vision components without additional complexity.

---

> > ### Author Rebuttal · Reviewer_9VTv · 2026-04-05
> >
> > My comments have been addressed. I will maintain my positive score.

---

### Decision · Program_Chairs · 2026-04-30

**Decision:**

Accept (regular)

**Comment:**

This paper focuses on catastrophic forgetting of multimodal large language models during downstream fine-tuning. It introduces Model‑Dowser, a data‑free sparse fine‑tuning approach that computes parameter importance scores by integrating output sensitivity, weight magnitude, and input activation via synthetic probing and efficient Jacobian estimation. By freezing high-importance parameters and updateing only low‑importance ones, the method mitigates forgetting while maintaining performance, as verified by experiments across LLaVA and NVILA models and different downstream tasks.

The paper is criticized by reviewers for limited novelty as its importance score only combines existing heuristics, insufficient strength of baselines, missing recent state-of-the-art comparisons, and inadequate presentation of key ablations. During rebuttal, the authors have clarified the novelty of their work, and provided more experimental evidences. Most reviewers have acknowledged concerns resolved. The AC agrees on the Accept recommendation. The authors should follow all the suggestions from reviewers in the revised version to strengthen the paper.